# The Spatial Pattern of Deprivations and Inequalities: The Case of Addis Ababa, Ethiopia

**Gizachew Berhanu Gelet** [1,*], **Solomon Mulugeta Woldemichael** [2] **and Ephrem Gebremariam Beyene** [3]

1 Ph.D. Program in Urban and Regional Planning, The Ethiopian Institute of Architecture, Building Construction and City Development (EiABC), Addis Ababa University, Lideta, Addis Ababa P.O. Box 518, Ethiopia

2 Department of Geography and Environmental Studies, Addis Ababa University, Addis Ababa P.O. Box 1176, Ethiopia

3 The Ethiopian Institute of Architecture, Building Construction and City Development (EiABC), Addis Ababa University, Lideta, Addis Ababa P.O. Box 518, Ethiopia

* Correspondence: gizber987@gmail.com; Tel.: +251-911-641-895

**Abstract:** Addis Ababa is a metropolitan area faced with the challenges of Ethiopia's urbanization, such as poverty, unemployment, informal settlements, an acute housing shortage, and environmental hazards. Yet, the non-practicality of area-based policy using the Multiple Deprivation Index (MDI) exacerbates the polarization of poverty and spatial inequality to create a divided city. The study developed the MDI for 2007 and 2016. The study's objective was to justify the area-based policy by analyzing the overlaps of deprivations based on the relationship of pertinent indicators and components, the spatial pattern of inequality and deprivations, and the relationship of deprivation with population size and density. The findings of the study were triangulated and validated with the deductive theoretical, empirical, and SDG frameworks to replicate external validity. The research design included both descriptive and correlational methods. The inductively derived pattern using PCA (principal component analysis) and LISA (local spatial association index) of MDI components revealed spatial inequality and poverty polarization. The index of concentrated poverty was revealed by global spatial autocorrelation. The statistical and spatial trend analysis revealed concentrated poverty, especially in the inner-city slums and the peri-urban informal settlements. Most of the findings conformed to deductive theoretical and SDG frameworks, while the analysis of MDI indicators and components revealed additional slum indicators and the relevance of integrating other SDG indicators with SDG 11 for realizing sustainable urbanization. Due to spatial inequality, patterns of concentrated poverty, a large, deprived population, and easing future urbanization challenges, the study rationalized area-based policy for reducing inequality and poverty polarization.

**Keywords:** multiple deprivation index (MDI); area-based targeting; spatial inequality; principal component analysis (PCA); spatial autocorrelation; SDG (sustainable development goal); concentrated poverty

## 1. Introduction

Development is not sustainable if people are excluded from opportunities, services, and a better life. Thus, SDG (Sustainable Development Goal)-10 calls for reducing inequalities in income, age, sex, disability, race, ethnicity, and economy [1]. Because of concentrated poverty, spatial inequality as measured by the Gini coefficient has not improved significantly in SSA (Sub-Saharan Africa), including Ethiopia, in recent years [2]. Ethiopia's urban inequality showed a slight increase from 0.29 in 1995 to 0.37 in 2010–11 and 0.38 in 2015–16, with more inequality in urban than rural areas [3–5]. Recent developments show that the income gap in Ethiopian cities is widening, with the bottom 10% of the population earning only 4% of the total income [6]. Increasing urbanization leads to higher income inequality because cities that produce more GDP attract more migrants from rural areas [7]. This is particularly true in the case of Addis Ababa, where rural-urban migration governs the

urbanization of Addis Ababa [8], and migrants account for 42 percent of its population [9]. Regardless of the persistent face of inequality, the Addis Ababa headcount poverty index was 16.8% in 2015/16, a significant improvement from 28.1% in 2010/11 [3,4]. Ethiopia's level of urbanization was 21.2 percent in 2019 [10] and is projected to reach 37 percent in 2035 [11], assuming that the country's urban growth rate is 5.4 percent per year [11,12].

In Ethiopia, urbanization has not been associated with a commensurate increase in economic prosperity. For instance, 23.5 percent of households (HHs) in Addis Ababa have recently reported the presence of unemployed adults [12]. To achieve middle-income status, Ethiopia needs to address the challenges of rapid urbanization, such as deepening poverty, high unemployment rates, the rapid expansion of informal settlements, an acute housing shortage, and the growing risk of environmental hazards [11]. Pro-poor spending in Ethiopia amounted to 65.7 percent of the total public expenditure in 2015/16 [4], while the reduction in non-monetary welfare (health, education, sanitation, and access to water) was at a low level [13]. Ethiopia should meet SDG target 1.2 of reducing by half the proportion of men, women, and children of all ages living in poverty in all its dimensions [14]. The MDI (Multiple Deprivation Index) is a means to link and monitor SDGs since it considers and weighs a multitude of problems. There is a link between SDG 4 on education, SDG 1 on poverty, and SDG 8 on employment. The above links are justified by the fact that education reduces poverty by increasing people's income and improving workers' productivity and productive capabilities. Yet, there is less evidence of the link between urban development (SDG 11) and education (SDG 4), except for the link between SDG 4 on education and SDG 11 on disaster management [15].

The development of MDI tools that monitor SDGs in an integrated manner and target resources to vulnerable, poor, and deprived areas will bridge the spatial inequality gap and reduce poverty's multidimensional problems. Nonetheless, the use of the MDI for resource allocation and prioritizing is less common in most countries in SSA. Addis Ababa allots a capital budget for sub-cities based on a sector-based approach that considers the unit cost of the project, population, and level of development [16] despite the large and disproportionally deprived population and enduring spatial inequality. In Ethiopia, the percentage of people living with multidimensional poverty (in terms of education, health, and living conditions) is 88% [17]. The non-monetary indicators [large HHs, high dependency rate, and lack of education] are the main characteristics of poor people in Ethiopia [18]. Urban inequality is rising in Ethiopia, from 0.29 in 1995 to 0.38 in 2016 [4]. As a result, the pragmatic application of MDI for disaggregated small neighborhood units (kebeles) reduces inequality and deprivations by implementing compensatory policies for targeted beneficiaries (i.e., in terms of poverty alleviation, the safety net, and social security), and prioritizing urban regeneration areas. MDI tools are useful for interpreting spatial inequality and developing a composite index for multidimensional deprivations. The MSAT (Multivariate Statistics Analysis Technique) is one of the empirical approaches to developing MDI that reflects the multifaceted nature of poverty faced by the urban poor. The MSAT's inductively derived pattern validation with the theoretical, empirical, and SDG frameworks enriches the local context pattern's replicabilities in other contexts and settings. The MSAT is a suitable instrument for refining the most crucial indicators among a multitude of deprivation indicators. As a result, it is useful for proper policy targeting, resource allocation, and figuring out the critical factors that explain concepts and theories [19–21]. Therefore, the first research question is stated as follows: "Which deprivation indicators are most strongly correlated with the main components of the MDI?" The main components of the MDI 2007 and 2016 explain the main factors to address in order to resolve the overlaps of deprivation and inequality experienced by the urban poor. The study used PCA (principal component analysis) over other MSAT approaches. This is because PCA is robust for maximizing the variance of variables and has other advantages detailed in the methodology sections of this work. The study interprets the inductively derived pattern of deprivations by analyzing the relationship between the crucial components and indicators of the MDI based on PCA using the SPSS (Statistical

Package for Social Science). The study discusses the rationale for area-based policy in light of the relationship between MDI indicators and components and the spatial patterns of the MDI components. The triangulation of the theoretical, empirical, and SDG frameworks validated and enriched the findings obtained regarding the interrelationships of indicators and components (Supplementary Materials).

Spatial inequality is caused by unequal distributions of income, resources, and infrastructure [22]. In Ethiopia, there is more inequality in urban than rural areas [5]. The poorest 10% of Ethiopia's population has not experienced income growth since 2005 [13,17], which substantiates the grave social inequality and social exclusion. Spatial inequality and poverty polarization are enduring facts since the destitute and vulnerable are concentrated in informal settlements and slums, while the rich are segregated into formal neighborhoods. Slums constituted 80 percent of the inner city of Addis Ababa [23]. In Addis Ababa, 66% of the population lives in informal settlements that cover about 44% of the city's built-up area [24]. Area-based targeting is rational for Addis Ababa, a city that is characterized by high rates of population, noticeable spatial inequality in access to good housing and basic urban services, as evidenced in its persistently high rates of unemployment, the worrisome incidence of poverty, and the conspicuous proliferation of informal settlements [11,12,25]. In 2017, Ethiopia spent 66.7% of its budget on anti-poverty interventions. The Urban Productive Safety Net Program (UPSNP) of Ethiopia has made some progress in the social inclusion of the poor, in improving their access to social services, and in enhancing their livelihood capital asset accumulations [26]. Nonetheless, the non-monetary dimensions of welfare, such as education, health, and access to water and sanitation, remain low [17]. Therefore, there is a need to align high pro-poor spending with spatially guided welfare-disadvantaged areas to know and prioritize locations to target resources for beneficiaries or perform urban regeneration interventions. In this regard, analyzing the spatial patterns of multidimensional factors or components is useful to prioritize strategic areas that need interventions to reduce spatial inequality and design policies to reduce the negative neighborhood effect of poverty concentration. Based on the preceding fact, the second research question is stipulated as follows: "Where are the highest and lowest deprivation concentration kebeles (neighborhood units) of Addis Ababa based on the spatial pattern of MDI components?" The study used the spatial autocorrelation tools of Moran's I to obtain an index for the city-wide deprivation concentration and the LISA (the local index of spatial association) to analyze the high and low deprivation concentration neighborhood units of Addis Ababa. Then, the study made policy recommendations based on its findings. The findings (for the first and second research questions) were triangulated for enriching internal validity, and then the triangulated findings are discussed in relation to theoretical, empirical, and SDG frameworks.

Area-based targeting offers completeness and efficiency in the case of the spatial concentration of poor individuals, reducing the negative neighborhood effect of poverty concentration through providing public goods and fund rationing [27,28]. The area-based approach supports the people-based approach to prioritize areas for extending economic resources and social protection measures to meet SDG 1 [29,30]. However, vulnerable people are numerous and disproportionally overcrowded in Addis Ababa's slum areas [24,31], resulting in a divided city. For Addis Ababa, the purely inner-city slum sub-cities accounted for 32.5% of the 2016 projected population [32], which was concentrated in 7.8% of the Addis Ababa area. The overall inner-city slum accounted for 40 percent of the population and 11 percent of the area of Addis Ababa [33]. In addition, the city periphery of Addis Ababa houses destitute and massive rural migrants, who acquire land through squatting or informal land transactions [8,34], while there are also some inner city squatters [31,35] and temporary and recent migrants in parts of the inner city slum [34]. Due to concentrated poverty, informal houses host a large and highly overcrowded poor population; as many as 35% of the residents of the inner-city slum live in single-room accommodations [36]. The current housing crisis in Addis Ababa is due to the escalating rural exodus to Addis Ababa [36]. Thus, rural-urban migration is the major factor driving urbanization [8] and

poverty concentration. Concentrated poverty is defined as the "spatial distribution of socio-economic deprivation", specifically focusing on the density of poor populations [37]. The theoretical frameworks substantiate the relationship between the pattern of concentrated poverty and population in the context of the global North [38–40]. The suburbanization of urban jobs and the exodus of middle-class blacks to white neighborhoods caused the concentration of underclass black people in American cities [38]. Alonso's bid rent model claimed that the poorest houses, poor people, and substandard buildings are concentrated on the outskirts of the city because the inner city is not affordable for the poor [39]. Moreover, the city center has a higher land value and population concentration than other areas, according to Alonso's model [40]. In the Ethiopian context, a link was established between urban forms and impoverished areas. Thus, concentrated poverty is higher in urban forms such as inner-city slums and peri-urban areas, while it is lesser in intermediate areas [41].

The empirical findings from the 122 World Bank poverty-targeted social programs showed that sector-based spending benefits the wealthy and that a quarter of these programs benefit non-poor people [42]. The preceding justifications point to the need for a policy framework, tools, and budget to better target vulnerable populations [29] living in high-poverty areas. In such cases, the MDI provides a tool to target and prioritize deprived populations. Currently, urban inequality and poverty polarization are rising in Ethiopia, which requires welfare-oriented strategies to better target compensatory budgets for vulnerable women and uneducated and impoverished HHs [5]. The analysis of locations with large, disproportionally deprived populations will assist in justifying the need for compensatory area-based policy to reach the more deprived populations, reduce spatial inequality, and address their preferences through participation and partnership. Given this fact, the third and fourth research questions address the relationship between deprivation and population size and density. The third and fourth research questions are stated, respectively, as follows: (3) What proportions of the sub-city population of Addis Ababa were most deprived? (4) Are there statistical correlations and spatial relationships between the MDI deprivation score and population density? The rationales for area-based policy are debated and justified by relating the findings of the above research questions to the theoretical, SDG, and empirical frameworks. The proportion of the most deprived population by sub-city is analyzed by descriptive statistics, and the statistical relationship between the deprivation score and population density is analyzed using PPMC (Pearson Product Moment Correlation). By comparing MDI 2007 and MDI 2016, the study interprets the spatial profile and trends of MDI deprivation scores as well as population density. The profile section stretches from the old inner city CBD (Central Business District) of the Addis Ketema sub-city to the Akaki Kaliti sub-city. Given that it was an old peri-urban informal settlement prior to the 2007 census, the profile trend extends to the Akaki-Kaliti sub-city fringe.

The study demonstrates how to generate a theoretical explanation for multiple deprivations and spatial inequality based on the pattern of relationships and overlaps of indicators and components derived in the inductive approach and triangulated with the spatial pattern of PCA components and deductive frameworks. The spatial inequality and poverty polarization in kebele (Addis Ababa's lowest administrative units), dominated by informal settlements, implied the need for area-based resource targeting for disaggregated small area units. The analysis of the deprivation patterns in line with theoretical, empirical, and SDG frameworks will enrich the external validity of the research in other contexts. Policymakers, planners, and multilateral and bilateral agencies can use the MDI for a range of applications. The MDI is useful for resource allocation, compensatory policy, tax exemption, and prioritizing poverty and social security beneficiaries. Furthermore, the MDI is used to promote community partnerships coupled with urban regeneration.

## 2. Evaluation of Multiple Deprivations and Policy Implications

### 2.1. Brief Overview of the Area and People-Based Policy Debate

There are debates on the pros and cons of people-based or place-based policies. Place-based policies are geographically targeted, with the intent and structure of helping disadvantaged residents in them. People-based policies help disadvantaged people without regard to where they live or how concentrated they are [43]. The goal of the program matters when adopting a place-based or people-based policy. If the poverty concentration is particularly pronounced (e.g., in the urban core), the location might help policymakers identify the intended beneficiaries. Nonetheless, if the goal is to improve access to low-income housing, a people-based program of vouchers is less wasteful and more targeted [28]. The 2009 World Bank report argued that governments should focus on economic concentration and people-based policies by providing universal welfare services at early stages of development for disadvantaged locations. Yet, spatial targeting is recommended for countries with high levels of urbanization, divided cities, large regional disparities, and small economies isolated from the world market [25].

Yet, area-based policies have a multitude of benefits from many perspectives [25,27,28,44–46]. The area-based policy makes sense for compensating areas with overlapping and coexisting problems, reducing spatial inequality, and reaching large, deprived populations who suffer disproportionally [44]. Area-based policies can help reduce residential segregation, bring about spatial and social justice, provide a framework for community planning and development, and reach some vulnerable sub-groups [25,27,45,46]. Residents of concentrated poverty frequently face more than limited individual resources. The provision of public goods (such as good education and crime reduction) has positive neighborhood and social network effects on poverty alleviation [28]. Area-based initiatives in degenerating urban areas foster the active participation of residents and the voluntary sector in England and Germany [45]. Area-based targeting has also been applied in SSA, especially in the urban planning experience of fast-growing cities and agglomeration economies [47,48]. Kinshasa, a city with a high rate of urban growth and agglomeration economies, planned spatially targeted priority areas and institutional and infrastructure improvements [47]. Addis Ababa implemented area-based targeting for prioritizing urban upgrading based on the criteria that an area should be targeted as an "upgrading area" in the statutory plan if a high proportion of its housing is lacking drainage and sanitation facilities [48].

### 2.2. MDI Tools and Area-Based Policy

Policymakers have used MDI tools to implement area-based or geographically targeted policies rather than sector-based budget allocation and uniform transfer budgeting based on population and other criteria. Using area-based policy, efficiency in allocating resources for poverty alleviation will increase, and leakage to the non-poor will be reduced [49]. The area-based policy is pragmatic by developing an MDI. The MDI is a relative measure of multiple deprivations at a small-area level and a tool used for allocating resources for poverty alleviation and urban regeneration [50,51]. Different countries have explored the MDI in different contexts by employing different methodologies [52–59]. The Welsh government has used the MDI for urban regeneration in partnership with the community [52,53]. The MDI has been used in England since 1990 to distribute renewal funds, stimulate the housing market, and provide tax exemptions [54]. The English MDI used factor analysis to give weight to factors and combine indicators [55]. The USA used the Alkire Foster method, which was done by counting and analyzing the different types of deprivations individuals experience and then deriving a multidimensional poverty index (MPI) to identify who is poor [56]. UN-Habitat designed a poverty alleviation program for SSA secondary cities based on multiple deprivation indicators [57]. South Africa prioritized social service delivery for disadvantaged groups using the MDI, developed based on PCA [58]. The India Slum Severity Index is applied to comprehend the extent of housing problems as well as to know the most deprived slum population [59].

SDG 1 calls for a policy framework for allocating a budget that disproportionately benefits deprived women, the poor, and vulnerable groups [29]. In the case of poverty polarization and fund scarcity, area-based approaches using MDI support a people-based approach for prioritizing vulnerable groups to meet SDG 1. The MDI is effective to locate vulnerable people (older people, people with disabilities, mothers, and the jobless) that require social protection benefits in line with SDG target 1.3 [30]. The MDI is also pragmatic to prioritize by area the poor and vulnerable people that require economic resources and service access, in line with SDG 1 target 1.4 [29]. In sum, the MDI is a crucial tool for targeting and prioritizing deprived and vulnerable groups, anti-poverty programs, identifying areas of less opportunity and resources, efficient allocation of resources, urban regeneration, fostering community partnership and participation, tax exemption, preventive health service delivery, and analyzing housing problems. The limitations of the MDI include missing a deprived population living in non-deprived areas, being less useful for rural areas due to dispersed spatial patterns, and taking some years to construct trend data on a sufficient number of MDI indicators. Several countries have started to use the MDI as a policy tool, which requires attention in the future on how to interpret and translate indicators into policy decisions [60]. Nonetheless, there is little experience in exploring the MDI for a range of applications in SSA, where the polarization of poverty is rampant, except in some countries in the southern parts of Africa.

### 2.3. Multiple Deprivations Concept and Indicators

Many empirical studies have identified a variety of monetary and non-monetary domains and indicators for explaining deprivations. Deprivation is defined as a lack of resources of all kinds and opportunities, while poverty is a lack of financial resources to meet needs [61,62]. The specified material deprivation variables for the Townsend index were unemployment, non-car ownership, non-home ownership, and overcrowding of HHs [61]. The 2019 English index of deprivation includes seven domains: income, employment, education, health and disability, crime, barriers to housing services, and deprivation of the living environment [55]. Multiple deprivations include non-monetary indicators such as overcrowding, insufficient water supply, poor sanitation, poor housing, limited access to education, inadequate protection of rights, being voiceless, and so on [63,64]. The income indicator takes precedence over non-monetary deprivations because a lack of income exposes the poor to non-monetary deprivations [65]. Since the two dimensions measure different kinds of deprivation, both the monetary and non-monetary dimensions must be taken into account while developing the MDI [66].

The empirical analysis revealed that the deprivation of economically vulnerable groups is multifaceted [67]. Deprivation is associated with social exclusion and vulnerable group indicators: FHHs [female-headed households], age, and disability [68–71]. The spatial organization theory linked deprivations to the spatial pattern of vulnerable groups, claiming that FHHs are particularly concentrated in the city's urban core [69]. Deprivation areas are also associated with morphological factors from remote sensing imagery, such as building density, building size, and green or open space [72,73]. SDG 11 target 11.7 calls for accessing green and public spaces for vulnerable groups (women, children, older people, and people with disabilities) [29]. Thus, the integrated analysis of morphology and social and economic vulnerability helps monitor SDG target 11.7.

Deprivation also varied in line with urban and settlement forms. Indicators of deprived green or open space, aged buildings, vulnerable communities, an overcrowded population, degenerating infrastructure, and dilapidated housing conditions characterize the inner-city slum of Addis Ababa [23,31,36,74]. Yet, Addis Ababa's peri-urban informal settlements are identified by the absence of tenure rights and a lack of infrastructure, consisting mainly of poor, vulnerable rural-urban migrants who acquired land through the transaction of agricultural land [8,31,34]. The SDG 11 housing inadequacy indicators [tenure rights, improved water and sanitation, housing durability, and adequate living space] are deprivation indicators specified by the SDG so that countries monitor their

progress in meeting the intended goals and targets [14]. The declining inner city, planned and new development regions, and peri-urban informal settlements were the emerging urban forms in Ethiopian urban centers [41], reflecting the relationship between urban form and deprived areas.

## 3. Materials and Methods

### 3.1. Description of the Study Area

The city administration of Addis Ababa was made up of 10 sub-cities and 99 kebeles when this study began (Figure 1). The inner-city slum of Addis Ababa covered 11% of the total area, covering the sub-cities of Lideta, Kirkos, Addis Ketema, Arada, and some parts of the Kolfekeranyo, Gulele, and Yeka sub-cities [31]. The area of Addis Ababa based on the 2007 census area delineation was 52,743 hectares. The population density of Addis Ababa was 160 and 190 people per hectare in 2007 and 2016, respectively [75]. The density is increasing despite urban renewal having displaced 28,584 HHs from 2009–2016 in Addis Ababa [76], while since 2012, the Ethiopian government has focused on the redevelopment of the inner city for the accumulation of high-end developers [33]. The inner city is mostly made up of old, unplanned, dilapidated, and kebele rental housing stock, though some have their own private tenure rights and some inner-city squatters exist [31,35]. The inner city of Addis Ababa lies 4.5 km from the city's main CBD, covering an area of 6050 hectares and housing 40% of the city's population [33].

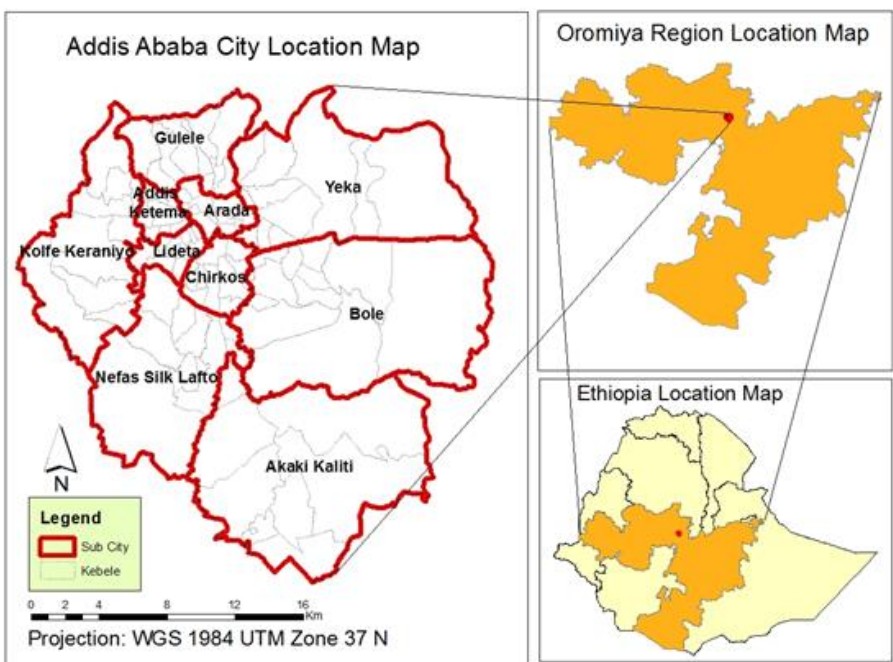

**Figure 1.** Sub-cities and Kebele boundary of Addis Ababa city.

The purely inner city consisted of 32 percent of the projected population [32], living in 7.8 percent of the area. The purely inner-city slum sub-cities are Lideta, Kirkos, Addis Ketema, and Arada. The old CBD is in the Addis Ababa inner-city market area of the Addis Ketema sub-city [77]. Merkato, situated in the old CBD, is the largest open market in Africa [78]. Based on a survey in wereda (a higher administrative unit next to a kebele) 07 of Addis Ketema sub-city, 39 percent of HHs rented beds for temporary and recent migrants [34]. The main expansion areas of squatter settlement are in the dominantly suburban sub-cities of Akaki Kaliti, Kolfe Keranyo, Yeka, and Bole [79]. The peri-urban informal settlement is an area subject to coercive bulldozing, little infrastructure, and uncertain or no tenure rights [31,35,41].

### 3.2. Socio-Economy and Other Characteristics of Addis Ababa as the Basis for MDI 2007

The population size of Addis Ababa, according to the results of the 2007 population and housing census, was 2,739,551, consisting of 628,985 housing units [80]. Housing tenure, including owner-occupied, rented, and rent-free housing, accounted for 32.6%, 61.45%, and 5.93% of the city's total housing units, respectively [80]. A total of 14.43% of the population aged five and older had never attended school. Migrants made up 47.6 percent of the total population. The migrant population for outside inner-city slum sub-cities was >50 percent for sub-cities (Bole, Nifas-Silk, and Kolfe-Keranyo) and 43 percent for the Akaki Kaliti sub-city. The disabled comprised 1.19% of the population [80]. The overall unemployment rate was 37.8% in 1999, 22.5% in 2007, and 21.2% in 2012 [80,81]. A total of 40 percent of the housing units had mud floors, 22.08% had no ceiling, and 98% had roofs made of corrugated iron sheets. For 76.89% of the housing units, the walls were built with mud and wood [80]. The average room number is 2.4 rooms per housing unit. Housing units with a tap inside the house and a tap in the compound constituted 5.83 percent and 25.89 percent, respectively. Housing with no sanitation facilities and housing with shared pit latrines constituted 14.3% and 41.1% of the housing stocks, respectively. A total of 86.25%, 40.79%, and 55.64% of the housing units have a radio, telephone, and TV, respectively [80]. Housing units with no bathroom and no kitchen room were 81.18% and 20.14%, respectively [80]. HHs that use electricity for cooking accounted for 34.71%. The average private and meter-shared electricity access in predominantly inner-slum sub-cities (Addis Ketema, Arada, Lideta, and Chirkos) was 99.2%, while the Addis Ababa average was 97.5% [80]. For the above-mentioned inner-slum sub-cities and Addis Ababa, the access to waste disposal services (including the municipality, private establishments, and individuals) was 85.1% and 69.6%, respectively [80]. About 70% of Addis Ababa's housing units were kebele and municipal rental houses, which were particularly concentrated in the inner city slums [31]. Based on the results of the 2018 survey conducted in the Addis Ababa case study area by the authors, 60% and 40% of the houses in the peri-urban squatter areas of Kolfe Keranyo sub-city and the inner-city slum area of Addis Ketema sub-city do not have a separate room for a kitchen, respectively.

### 3.3. Socio-Economic and Other Characteristics of Addis Ababa as the Basis for MDI 2016

The projected population size for 2016 was 3,352,000 [32]. Based on the SPSS 20 analysis of the 2015/16 HH expenditure survey data from the CSA (Central Statistics Agency of Ethiopia), the number of HHs was 3832 (44.02% of the HHs were female) in Addis Ababa. FHHs, with a widowed or divorced marital status, constituted 23.05 percent of the HHs. The disabled constituted 2.46 percent of the HHs. Unemployed and illiterate HHs made up 22.6% and 15.94% of the total HHs, respectively. HHs with a bachelor's degree or higher constituted 3.92 percent. Those aged 65 and above constituted 15.26% of the total HHs. Out of the HHs, 11.93% engaged in formal self-employment. Service workers and shop market sales accounted for 39.4% of self-employed formal businesses, followed by elementary (35.2%), craft-related (13.6%), and the remaining (11.8%) [4]. Based on the existing land use of Addis Ababa in 2017, the road and green area constituted 10 percent and 34.90 percent, respectively. Building footprint areas, calculated using a 2011 aerial photograph, covered 11% of Addis Ababa.

### 3.4. Methodological Procedure

CSA is the official data provider for monitoring and evaluation tools [3,4]. Ethiopia conducted censuses in 1984, 1994, and 2007. The government indefinitely postponed the 2018 census due to social unrest. The MDI 2007 used the population and housing census for 99 kebeles of Addis Ababa. The MDI 2016 used the 2016 CSA HH expenditure and socio-economic survey, the 2016 CSA population projection [32], the base map of the 2017 structure plan of Addis Ababa, the building footprints of the 2011 Addis Ababa aerial photograph, and the 2016 CSA population projection of Addis Ababa. The 2016 CSA HH expenditure survey covered 93 kebeles out of 99 kebeles in Addis Ababa. The requisite data

for the remaining six kebeles were estimated based on the average values of the surveyed kebeles that surrounded them.

The descriptive research design uses percentages, maps, or graphs to describe the situation. Moreover, the author's previous research on case studies of peri-urban and squatter settlement areas of Addis Ababa [34], supported by the corresponding image interpretation for morphology and physical observation, enriched the interpretation and discussion of the findings. The correlational research design analyzes the relationship between multiple or two variables using the analytical methodologies of PCA, PPMC (Pearson Product Moment Correlation), LISA, and Moran's I. The unitary weighting method and asking for opinions are simple but subject to arbitrary and subjective judgment [19]. The MSAT statistical weighting and ranking deprivations based on component (factor) scores provide an advantage relative to the non-statistical weighting method. In MSAT statistical weighting, the obtained factors clarify the general concept via an empirical link among a set of indicators, which makes the MSAT method appropriate for policy targeting [19–21]. MSAT's inductive approach derives indicators, components, and patterns from observation and the development of explanations (theories) through a series of hypotheses [21]. PCA overrides other MSAT methods for the MDI model. The benefits of PCA are an orthogonal transformation of the original variables into a new set of variables, maximizing the variance of variables, and reducing redundancy. In addition, PCA derives small diagnostic factors from a large set of variables, and it is advantageous for giving more weight to unequally distributed assets between cases [82–85]. The study joins PCA results in SPSS 20 with the Kebele spatial unit of AA in ArcGIS 10.8 for GIS-based analysis. The overall methodological flows and procedures are illustrated in Figures 2 and 3. The methodological procedures are specified in a step-wise manner as follows:

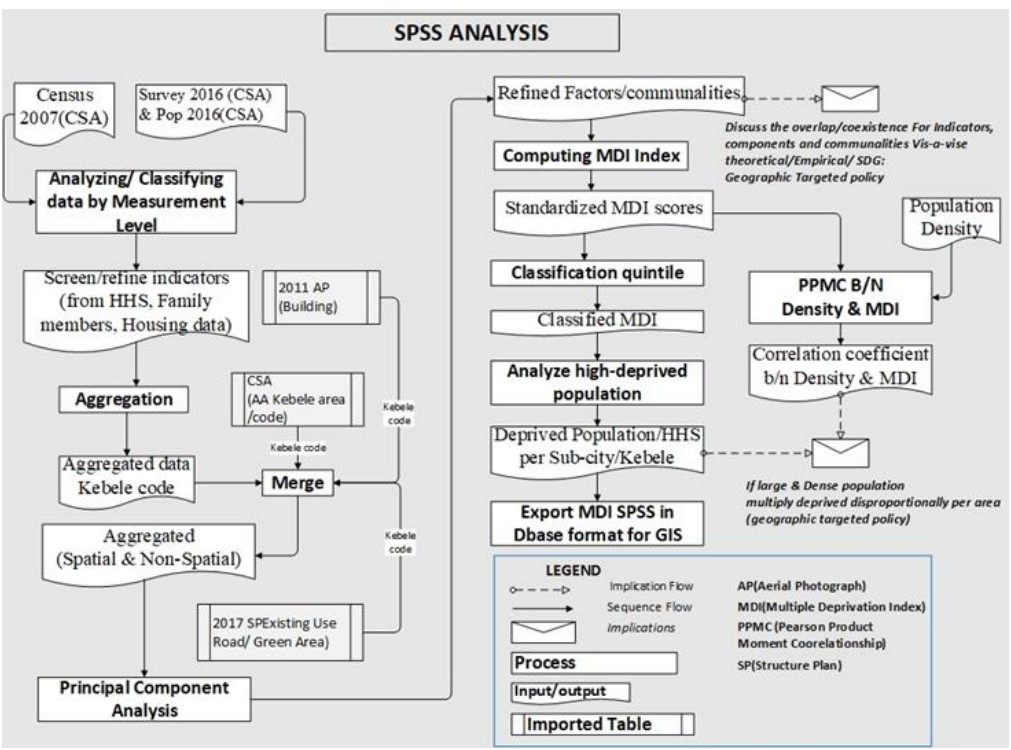

**Figure 2.** Methodological framework SPSS analysis.

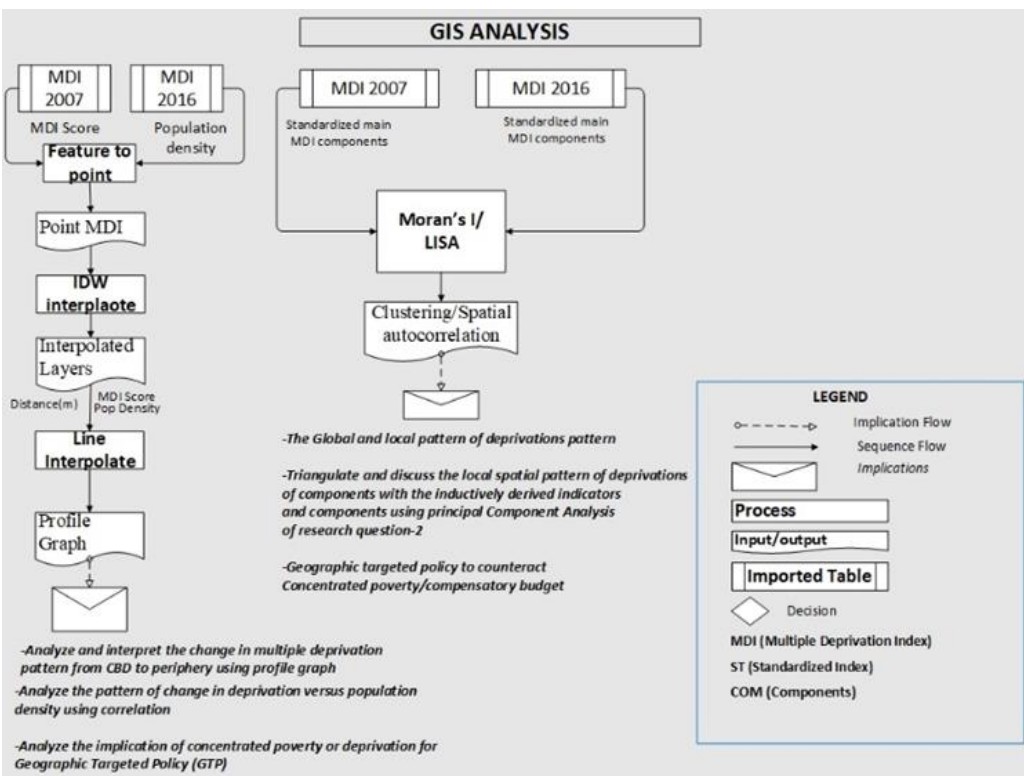

**Figure 3.** Methodological framework GIS analysis.

Step-1 Variable Selection and description

The SPSS correlation analysis refined 23 and 11 factors for MDI 2007 and MDI 2016, respectively. Kebele was the disaggregation unit for population and HH-level data. By screening variables, which measure the concept or construct, through the deductive theoretical lenses and the inductive empirical approach of MSAT, the research enhanced the internal validity of the variables. The SPSS correlation analysis refined 23 and 11 indicators or variables for MDI 2007 and MDI 2016, respectively. Kebele was the disaggregation unit for population and HH-level data. By screening indicators that describe the concept or construct through the deductive theoretical lenses and the inductive empirical approach of MSAT, the research enhanced the internal validity of the variables. The findings were triangulated within the theory, analysis, and SDG contexts to ensure external validity. The dimensions and indicators for MDI 2007 and MDI 2016 are described in Table 1.

Step 2: Assessing variables' appropriateness and factorability

Spearman's rank correlation coefficient (SMRCC) was used to screen candidate variables for PCA analysis. For MDI 2007 and MDI 2016, the factorial ecology model of PCA was applied in SPSS 20 to the screened variables by SMRCC. The PCA tool performed the orthogonal transformation of the observed variables (screened variables) into a new set of variables. Keyser–Meyer–Olkin (KMO) sampling adequacy tests the suitability of the data for PCA analysis. Due to the KMO being greater than 0.5 when comparing observed and partial correlation [86,87], the PCA analysis of indicators and components proceeded. The Bartlett test of sphericity (BTS) measures the factorability of the inter-correlation matrix. The BTS was less than 0.05, which implies that the null hypothesis that the correlation matrix is uncorrelated was rejected.

**Table 1.** Dimensions/factors, indicators, and description for MDI 2007 and MDI 2016.

| Factors MDI 2007 | Indicators | MDI Year | Indicator Description |
|---|---|---|---|
| Education | Illiterate population | MDI 2007 | % of illiterate family members of the HHs |
| | Degree level education deprived | MDI 2007 | % family members of the HHs with education < bachelor's degree level |
| Housing condition | Wall deprived | MDI 2007 | % of housing with substandard wall materials (mud, wood, thatch, stones, and their combinations) |
| | Floor deprived | MDI 2007 | % of housing with sub-standard floors (mud, bamboo reed, and their combinations) |
| | Ceiling deprived | MDI 2007 | % of housing without ceiling |
| | Aged house | MDI 2007 | % of housing age $\geq$ 20 years |
| | Deprived housing facilities | MDI 2007 | % of housing-deprived of three facilities (radio, TV, house phone). For example, 0% deprived means HHs have three facilities, and 100% deprived means HHs have zero (no) facilities. |
| Health | Over crowdedness | MDI 2007 | % of HHs over-crowdedness $\geq$ 2.5 person/room |
| | Population density | MDI 2007 | 2007 census population per kebele area in hectares) |
| Services and infrastructure | Own piped water-deprived | MDI 2007 | % of housing using shared pipe tab, protected/unprotected well, river/lake/pond |
| | Sanitation-deprived | MDI 2007 | % of housing with either a pit latrine shared by one or more HHs or no toilet at all |
| | Waste disposal-deprived | MDI 2007 | % of housing that was not using public dump/private house collection |
| | Modern cooking-deprived | MDI 2007 | % of housing which was not using gas and electricity for cooking |
| | Electric light-deprived | MDI 2007 | % of housing that was not using electricity for lighting |
| | Bathing facility proportion | MDI 2007 | % of housing with no bath facilities |
| | Kitchen-deprived | MDI 2007 | % of housing with no specific kitchen room |
| Tenure | Private tenure owner deprivations | MDI 2007 | % of housing whose occupant is not the owner of the house (the occupant is a renter or rent-free) |
| Social Vulnerability | Disabled population | MDI 2007 | % of disability per population |
| | Migrant population | MDI 2007 | % of migrants per population |
| | Widowed/divorced FHHs | MDI 2007 | % of FHHs with marital status widowed or divorced |
| | Unemployment | MDI 2007 | % of the unemployed population with age $\geq$10 years and <65 years per economically active population (10–64 age group based on Ethiopian census) |
| | Old dependency rate | MDI 2007 | % of the population $\geq$ 65 years old per productive force population (15–64 age group) |
| | Young dependency rate | MDI 2007 | % of the population with 0–14 years old per productive force population (15–64 age group) |

| Factors MDI 2016 | Indicators/Variables | MDI Year | Indicator Description |
|---|---|---|---|
| Income | 2016 Income per AE | MDI 2016 | HH income per adult equivalent (AE) per month for 2016. AE indicates poverty measure is adjusted for the difference in the calorie requirements of different HH members (for age and gender of adult members). |
| Education | Illiterate HHs | MDI 2016 | % HHs with illiterate educational status |
| | Non-degree HHs | MDI 2016 | % HHs not having degree and above education status |
| Employment | Self-employed HHs | MDI 2016 | % HHs employed in self-employed formal business. |
| | Unemployed HHs | MDI 2016 | % HHs unemployed from economically active age group (10–64) |
| Social vulnerability | FHHs widowed/ Divorced | MDI 2016 | % FHHs with widowed/divorced marital status |
| | Older HHs | MDI 2016 | % HHs with age 65 and above |
| Overcrowding | Population density | MDI 2016 | 2016 projected population per kebele area in hectares |
| | Building density | MDI 2016 | Building footprint area percentage per kebele (based on 2011 15 cm aerial photograph) |
| Environment | Green per capita | MDI 2016 | Green area coverage in meter square per 1000 population of kebele (based on existing land use prepared for the 2017 structure plan of Addis Ababa) |
| Infrastructure | Road per capita | MDI 2016 | Road area coverage in hectares per 1000 population of kebele (based on existing road area for the 2017 structure plan of Addis Ababa) |

Step-3: Refining Explanatory components

PCA is an orthogonal transformation of a system of variables $x_1, x_2, \ldots, x_n$, into $y_1, y_2, \ldots, y_p$. It has a mathematical form:

$$x_i = b_{i1}y_1 + b_{i2}y_2 + b_{i3}y_3 + \ldots + b_{ik}y_k$$
$$i = 1, 2, \ldots, n$$
$$k = 1, 2, \ldots, p, \qquad (1)$$

The number of empirical variables is equal to the number of components ($n = p$), and the total variance of the variable $x_i$ is equal to the component variance $y_k$ [88]. The PCA varimax rotation maximized the sum of the variances of the square loadings. Because of the rotation, each component has a small number of higher loadings, simplifying items' loadings by removing the middle ground [89]. Components with an Eigenvalue of greater than one indicated significant components, which explained most of the variance in the original data set. Component loading is the correlation between a specific observed variable and a specific component. Communality is the sum of all the squared factor loadings, and it is the same as $r^2$ in regression analysis [90]. The SPSS output of the rotated correlation matrix skipped the variable's loading score between −3 and 3. The PCA interpretation skipped variables with a communality score of less than 0.3 due to the particular variable's weak relationship to a particular principal component [91]. The direction and strength of the relationships between the component's indicators, with high loading scores, were taken into consideration when naming the components.

Step-4: Develop a Non-Standardized Index (NSIMD) and a Standardized Index (MDI) of Multiple Deprivations

There are two MDI construction scenarios. Regarding the first scenario, only the first component that loaded highly on many variables was a measure of multiple deprivations [21]. The second scenario, the one used in this research, first developed NSMDI (non-standardized MDI), based on the percent of variance explained by components with eigenvalues > 1, the total variance explained by considered components for the MDI, and the component score for each unit (Kebele) [86]. Secondly, the standardized MDI was developed by standardizing the NSMDI with a linear function between the absolute highest and lowest score [86,92]. Accordingly, the MDI summarizes complex dimensions of deprivation into a single, easy-to-use numeric representation [86,93]. The NSMDI formula is specified below.

$$NSMDI_t = \left(\frac{c1}{cs}\right) \times (f1) + \left(\frac{c2}{cs}\right) \times (f2) + \left(\frac{c3}{cs}\right) \times (f3) + \ldots \left(\frac{cn}{cs}\right) \times (fn) \qquad (2)$$

The $NSMDI_t$ indicates NSMDI at time $t$, and $t$ designates the agreed-upon MDI year setting. The designation code, *c1* up to *cn*, denotes the variance explained by each component with an eigenvalue > 1, where ci ranges from *c1* to *cn*. *f1* to *fn* are the component scores of each kebele unit of Addis Ababa, and fi varies from *f1* to *fn*. cs is the sum of the variance explained by the components (*c1* to *cn*). The MDI formula is specified below.

$$MDI_t = \left| \frac{(NSC - NLV)}{(NHV - NLV)} \right| \times 100 \qquad (3)$$

The $MDI_t$ designates MDI at time $t$ (year). The designation $NSC$ denotes the $NSMDI_t$ coefficient value of each kebele unit of AA. $NLV$ denotes a low coefficient value for $NSMDI_t$, while $NHV$ denotes a high coefficient value of $NSMDI_t$. The standardized $MDI_t$ ranged from 0% (no deprivation) to 100% (maximum deprivation).

Step-5: MDI classification and analysis of the proportion of the "most deprived population".

The MDI is classified into quintiles using an equal-interval approach. The MDI ranged from a standardized score of 0% to 25% (low-deprived, or LD), 25–50% (deprived, or

DE), 50–75% (high-deprived, or HD), and 75–100% (very high-deprived, or VHD). "Most deprived" comprised the HD and VHD.

Step-6: Preparing profile graph for deprivations/population density

ArcGIS software converted the MDI polygon layers to point layers and ran interpolation on the point layers using the Inverse Distance Weighting tool. The profile graph was derived in ArcGIS using the interpolated raster as the background layer. The profile graph modeled the spatial trend in deprivation score and population density as distance increased from the CBD to the periphery. The profile graph plotted the 20 km cross-section spanning from the old CBD towards the outskirts of the Akaki Kaliti sub-city of Addis Ababa. The profile graph pointed from the CBD to the Akakai Kaliti sub-city periphery (rather than other city fringe areas), as the Akaki Kaliti sub-city had peri-urban informal settlements before the 2007 census.

Step 7: Analyzing the spatial pattern of multiple deprivations: Moran's I index and LISA (Local Spatial Autocorrelation)

"Moran's I" is an index used to indicate the degree of spatial polarization of deprivations for this study. When similar standardized MDI component values cluster together, the index is positive, while the index is negative when dissimilar values cluster together. The MDI component value close to "0" indicates no spatial autocorrelation. Moran's I give the overall spatial autocorrelation, but it is not useful to distinguish the variation in local spatial patterns of the MDI components [94]. Moran's I statistic ($I$) for spatial autocorrelation and $Z_I$ score ($Z_I$) for the statistics [94] are given as follows:

$$I = \frac{n}{s_o} \frac{\sum_{i=1}^{n} \sum_{j=1}^{n} wi,j^{z_i z_j}}{\sum_{i=1}^{n} zi^2} \qquad S_o = \sum_{i=1}^{n} \sum_{j=1}^{n} wi,\ j$$

$$Z_I = \frac{I - E[I]}{\sqrt{V[I]}} \qquad \text{Where}: \ E[I] = {}^{-1}/(n-1)$$

$$v(I) = E(I^2) - E(I^2) \tag{4}$$

Based on Moran's I statistics ($I$), $z_i$ is the x-component value of the kebele "i" deviation from the mean value ($x_i$-$\overline{x}$). $Wi,j$ is the spatial weight between kebeles "$i$" and "$j$", and $n$ is the total number of kebeles (neighborhood units). $s_o$ is the aggregate of all spatial weights. When the *p*-value is statistically significant, for the component with a positive or negative z-score, you may reject the null hypothesis.

LISA is used to assess local hotspots and the impact of individual locations on the magnitude of "Moran's I" and to identify outliers [95]. A positive value for LISA indicates spatial clustering of similar values (either high or low values), and a negative value indicates spatial clustering of dissimilar values. In this paper, LISA examines the local-specific pattern of a given MDI component value that most strongly contributes to spatial inequality and poverty polarization. GeoDa software computed LISA and Moran's I spatial statistics, considering the weight given for each "kebele," or neighborhood unit, using queen contiguity weight. The LISA formula is specified as indicated below, based on Anselin (1995) [95].

$$I_i = z_i \sum_j w_{ij\ z_{j,}} \tag{5}$$

Analogous to the global Moran's I, the observation $z_i$, and $z_j$ are in deviation from the mean. The summation over j is such that only neighboring values j $\in$ $J_i$ are included. $w_{ij}$ denotes weights, which may be in the row-standardized form or not. The weight $w_{ij}$ is, by convention, equal to "0" [95,96]. A high-high (H-H) LISA relation indicates that a neighborhood with a significantly higher value is surrounded by neighborhoods with higher values, while a lower neighborhood value is surrounded by a lower neighborhood value for a LISA low-low (L-L) relation. A high-low (H-L) LISA relationship indicates that a neighborhood with a higher value is surrounded by neighborhoods with lower values, while a lower neighborhood value is surrounded by neighborhoods with higher neighborhood values for a low-high (L-H) relationship.

The PCA pattern is triangulated with the LISA pattern to augment the internal validity of the findings. The LISA interpretations were further synchronized with formal and informal settlement morphologies from a Quickbird image of 2009 (for MDI 2007) and a historical Google image from 2016 (for MDI 2016). In addition, our previous study and ground verifications of informal settlements enriched the interpretations of the overall findings. By interpreting low (LISA L-L) and high (LISA H-H) deprivation clusters, areas of polarization of poverty and spatial inequality were identified for each PCA component.

## 4. Results

### 4.1. PCA Result for MDI 2007

SPSS 20′s correlation matrix identified 23 indicators or variables with significant correlations ($p < 0.05$). The overall collinearity was 5.108 (no-collinearity), exceeding the determinant threshold ($p > 0.00001$). The KMO was 0.842, which implied the sample size was sufficient to proceed with factor and PCA analysis. The BTS was significant ($p < 0.001$), and the Chi-square ($X^2$) was 3151.339 to reject the null hypothesis that the correlation matrix is uncorrelated. The communality for all 23 variables in the 2007 census was greater than 0.3. For further reference on indicators, components, and communalities, see Table 2. The communality ($r^2$), or percent of variance explained by the model, was >0.6 for all 23 indicators of the MDI 2007. The first, second, third, and fourth components explained 40.25%, 26.97%, 9.218%, and 4.783% of the variation in the original data set, respectively. The communality is >0.6 for all indicators and >0.89 for nine (9) indicators. See Table 2 for a further overview of the 4 components and 23 indicators.

**Table 2.** Rotated Component Matrix of PCA for 24 deprivation indicators and 4 components/factors based on the 2007 census for Addis Ababa City.

| Indicators | Communality | Component 1 | Component 2 | Component 3 | Component 4 |
|---|---|---|---|---|---|
|  |  | LHSS | HEPSV | VUSG | PHSC |
| Illiterate population | 0.899 | 0.783 |  | 0.438 | 0.301 |
| Degree-level education-deprived | 0.763 | 0.826 |  |  |  |
| Wall-deprived | 0.817 | 0.566 | 0.586 | 0.388 |  |
| Floor-deprived | 0.930 | 0.928 |  |  |  |
| Ceiling-deprived | 0.922 | 0.819 | −0.484 |  |  |
| Aged houses | 0.935 | −0.321 | 0.703 | 0.566 |  |
| Deprived housing facilities | 0.935 | 0.938 |  |  |  |
| Over crowdedness | 0.855 | 0.305 | 0.85 |  |  |
| Population density | 0.730 |  | 0.759 |  | 0.324 |
| Own piped-water-deprived | 0.667 | 0.809 |  |  |  |
| Sanitation-deprived | 0.763 | 0.829 |  |  |  |
| Waste disposal-deprived | 0.851 | 0.796 | −0.427 |  |  |
| Modern cooking-deprived | 0.869 | 0.886 |  |  |  |
| Electric-light-deprived | 0.925 | 0.59 | −0.520 | 0.359 | 0.411 |
| Bathing facility proportion | 0.598 |  |  | −0.714 |  |
| Kitchen-deprived | 0.788 |  | 0.376 |  | 0.786 |
| Private tenure owner deprivations | 0.919 | −0.471 | 0.812 |  |  |
| Disabled population | 0.349 |  | 0.484 |  |  |
| Migrant population | 0.869 | 0.415 |  | 0.727 | 0.317 |
| Widowed/divorced FHHs | 0.889 |  | 0.670 | 0.620 |  |
| Unemployment | 0.613 |  | 0.768 |  |  |
| Old dependency rate | 0.890 |  |  | 0.897 |  |
| Young dependency rate | 0.904 | 0.767 | −0.530 |  |  |

Component 1: It is named Low Human Capital and Substandard Services (LHSS), which is based on the strength of this component's indicators' loading scores and the direction of relationships. In generic form, the loading score (>0.76) is higher for three indicators describing low human capital asset proportions (higher illiteracy, deprivation of a high education level, and a higher proportion of young dependents). This component has a high loading score (>0.59) in generic form for substandard housing materials, poor housing facilities, poor services (sanitation, water, modern cooking), less electricity provision, and a low proportion of waste disposal services. In sum, this component's physical deprivation and low human capital are crucial aspects.

Component 2: It is named Health, Social, and Physical Vulnerability (HEPSV) based on the strength of the loading score (>0.48) for three indicators that contribute to public health deterioration (higher overcrowding, higher population density, and higher disability). The high loading score (>0.66) for the indicators (unemployment and FHH widows or divorced) demonstrates the area's social and economic vulnerability. The area's physical assets and private tenure ownership deprivations were implicated by the high loading score (>0.7) for indicators (old-aged houses and private tenure owner deprivations). HEPSV loading scores, on the other hand, describe a lower deprivation of electric light and waste disposal services (−0.4), which is typical in the inner-city area. The lower percentage of young dependents (−0.53) relative to other components means the area has an older population.

Component 3: It is named Vulnerable Social Groups (VUSG), primarily considering the high loading score (>0.72) for indicators of vulnerable groups (a higher proportion of old dependents and a higher proportion of migrants). VUSG's loading score (−0.71) explains the low percentage of bathing facilities. Furthermore, this component ranks second, relative to other components, regarding the proportion of FHH widows and divorced, illiterate HHs, aged houses, and lack of access to electricity.

Component 4: This component is known as "poor housing services and congestion" (PHSC) due to a high loading score (>0.4) of low physical capital assets (no specific kitchen room in the house and a high proportion of deprived electric lighting). It also indicated a high loading score (>0.3) for high population density next to the second component. The non-provision of electrical services is a peri-urban or fringe neighborhood settlement feature. This is due to the fact that the government did not install electricity on undeveloped land in accordance with the statutory plan. The deprivation of the kitchen reflects the features of inner-city slums and peri-urban informal settlements.

### 4.2. PCA Result for MDI 2016

For MDI 2016, the correlation matrix in SPSS 20 screened 11 variables with significant correlation ($p < 0.05$). The overall collinearity was 0.005, exceeding the determinant threshold ($p > 0.00001$). The KMO was 0.652, which assured that factorial analysis could proceed for MDI 2016. The BTS was significant ($p < 0.001$), and the Chi-square ($X^2$) was 461.879, significant enough to reject the null hypothesis that the correlation matrix is uncorrelated. The variations explained by the first, second, and third factors/components are 27.44%, 20.620%, and 11.87%, respectively. The communality for all MDI 2016 indicators is greater than 0.6. See Table 3 for a further overview of the three components and eleven indicators.

Component-1: It is named "Congested Living and Vulnerable Social Group (CLVS)" considering the high positive loading score for indicators of this component. The high loading score (>0.47) for widowed or divorced FHHs and unemployed HHs explains the high proportion of vulnerable groups. The high loading score (>0.58) for buildings and population density also signified substandard housing and a lack of living space for vulnerable groups. The overcrowded living conditions and concentration of vulnerable groups depicted Addis Ababa's inner-city areas.

**Table 3.** Rotated Component Matrix of PCA for 11 deprivation indicators and 3 components of MDI 2016.

| Indicators | Communality | Component-1 | Component-2 | Component-3 |
| --- | --- | --- | --- | --- |
| | | CLVS | LSES | EVIN |
| Illiterate HHs | 0.687 | 0.299 | 0.703 | 0.323 |
| Non-degree HHs | 0.425 | −0.312 | 0.572 | −0.010 |
| Self-employed HHs | 0.600 | −0.020 | 0.774 | −0.024 |
| Unemployed HHs | 0.263 | 0.478 | −0.168 | −0.082 |
| 2016 income per AE | 0.548 | −0.096 | −0.719 | −0.147 |
| Widowed/divorced FHHs | 0.643 | 0.777 | 0.196 | −0.027 |
| Older HHs | 0.607 | 0.779 | 0.009 | 0.018 |
| Building density | 0.781 | 0.660 | 0.232 | −0.540 |
| Population density | 0.702 | 0.584 | 0.371 | −0.473 |
| Green per capita | 0.807 | 0.012 | 0.097 | 0.893 |
| Road per capita | 0.906 | −0.183 | 0.076 | 0.931 |

Component 2: This component is called "Low Socio-Economic Status (LSES)". The higher positive loading scores (>0.57) for the percentage of non-degree HHs, illiterate HHs, and self-employed HHs are interwoven to form low human capital assets. On the contrary, there is an inverse relationship between low income (−0.719) and a high proportion of illiterate people (0.687), a low proportion of degree-level HHs (0.425), and a high proportion of low-paid self-employed jobs (0.6). As a result, low income and low human capital assets result in low socioeconomic status. The preceding statement indicates that low education leads to low-income and low-paid jobs that, in turn, enforce poverty traps for individual HHs and neighborhood HHs, which seeks further research on the negative neighborhood effects of poverty concentration.

Component 3: The component is named Environment and Infrastructure (EVIN) based on the high loading score for green space and roads per capita. The EVIN component revealed an inverse relationship between a high proportion of roads per capita (0.931) and green per capita (0.893) on the one hand and a low proportion of the population density (−0.473) and building density (−0.54) on the other hand. In general, the amount of green space and infrastructure per capita is low in overcrowded slum areas, while it is high in newly developed, formal suburban areas.

*4.3. Computing Non-Standardized MDI (NSMDI) and Standardized MDI (SMDI)*

The NSMDI and MDI were computed for MDI 2007 and MDI 2016. The four components of MDI 2007 and the three components of MDI 2016 explained the overall variation of 81.2 percent and 59.93 percent of the original datasets, respectively. The NSMDI and MDI were computed for 2007 and 2016 for components with Eigen values >1.

$$NSMDI_{2007} = \left(\frac{40.25}{81.22}\right) \times (f1) + \left(\frac{26.97}{81.22}\right) \times (f2) + \left(\frac{9.22}{81.22}\right) \times (f3) + \left(\frac{4.78}{81.22}\right) \times (f4)$$

$$NSMDI_{2016} = \left(\frac{27.44}{59.93}\right) \times (f1) + \left(\frac{20.62}{59.93}\right) \times (f2) + \left(\frac{11.87}{59.93}\right) \times (f3)$$

The MDI was computed for 2007 ($MDI_{2007}$) and 2016 ($MDI_{2016}$) based on the formula below.

$$MDI_{2007} = \left|\frac{(NSC-NLV)}{(NHV-NLV)}\right| \times 100 = \left|\frac{(NSC-(-1.8982))}{(1.3954-(-1.8982))}\right| \times 100$$

$$MDI_{2016} = \left|\frac{(NSC-NLV)}{(NHV-NLV)}\right| \times 100 = \left|\frac{(NSC-(-1.3101))}{(1.5547-(-1.3101))}\right| \times 100$$

*4.4. The Overall Spatial Pattern of Deprivations Concentration*

For the MDI 2007 classification, the most deprived quarters were pervasive in most parts of the city, while the intensity of deprivation was less for most of the intermediate areas. Yet, for the MDI 2016 classification, the most deprived areas showed vivid clustering

in the inner-city slums and peri-urban informal settlements, while the intermediate and formal suburban areas were less deprived quarters. See Figure 4 for MDI 2007 and MDI 2016 deprivation classifications.

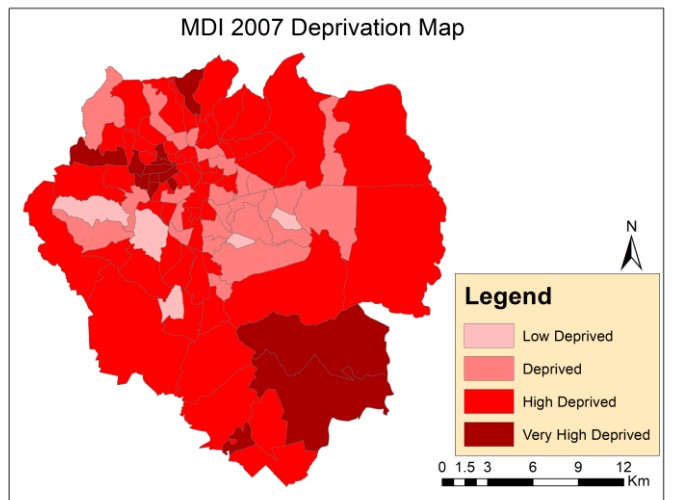 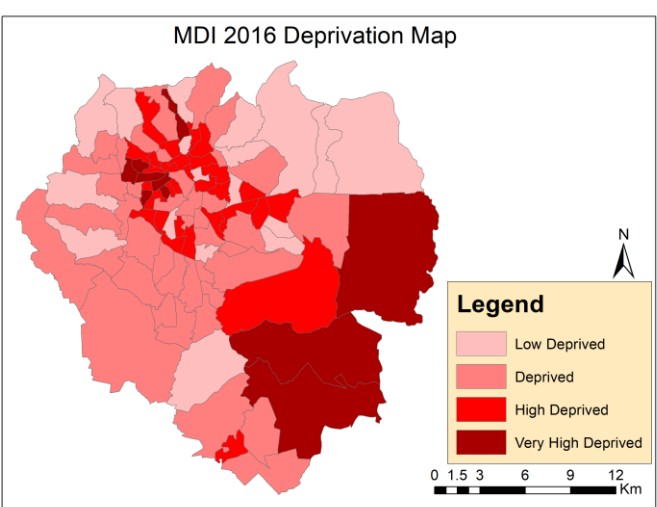

**Figure 4.** Deprivation classification for MDI 2007 and MDI 2016.

For components 2007 and 2016, the result of Moran's I indicated significant positive spatial autocorrelation, while the strength of the "Moran's I" coefficient was high for MDI 2007, relative to MDI 2016, due to the consideration of a multitude of census indicators. The global "Moran's I" for MDI 2007 and MDI 2016 indicated that neighborhood units with high deprivation scores were close to each other, revealing the spillover effect of poverty concentration on adjacent neighborhoods. Component 2 (HEPSV) of the MDI 2007 demonstrated strong positive spatial autocorrelation, relative to other MDI 2007 components, justifying how the deterioration of public health and vulnerability formulated a concentrated poverty pattern and spatial inequality. For MDI 2016, Moran's I result was positive and strong for component 3 (EVIN), compared to other MDI 2016 components, verifying how a lack of road infrastructure and green space per capita leads to a pattern of concentrated poverty. See Table 4 for the result of the global "Moran's I" index.

**Table 4.** Moran's index for components of MDI 2007 and MDI 2016.

| Component (2007) | Component 1 (LHSS) | Component 2 (HEPSV) | Component 3 (VUSG) | Component-4 (PHSC) |
|---|---|---|---|---|
| Moran's I (2007) | 0.463 | 0.604 | 0.478 | 0.469 |
| Component (2016) | Component-1 (CLVS) | Component-2 (LSES) | Component-3 (EVIN) | |
| Moran's I (2016) | 0.363 | 0.283 | 0.445 | |

### 4.4.1. The Local Spatial Pattern of Deprivations and Inequality for MDI 2007

The LISA H-H (red color) showed high deprivation in inner-city slums as well as informal settlements (peri-urban and suburban areas) for MDI 2007, while the blue color indicated low deprivation clusters (LISA L-L). The light blue on the LISA indicated low-high (L-H), while the light red indicated high-low (H-L).

*MDI 2007 Component-1 (LHSS) spatial pattern*

The LISA H-H relation indicated a high level of deprivation in the suburbs and peri-urban areas of the Bole and Akaki Kaliti sub-cities. In conjunction with this component PCA pattern, the high deprivation spatial pattern suggested a lack of durable housing materials, environmental services, and human capital assets in the outskirt informal settlements. Low deprivation was identified by the LISA L-L for areas that were dominated by formal settlements (the sub-city areas in the eastern parts of Chirkos, parts of Yeka, and the

northern part of Bole). Hence, the LISA L-L showed that the Arada sub-city, which served as Addis Ababa's former cultural and educational hub, had a low concentration of deprivations. Look at Figure 5 for a further overview.

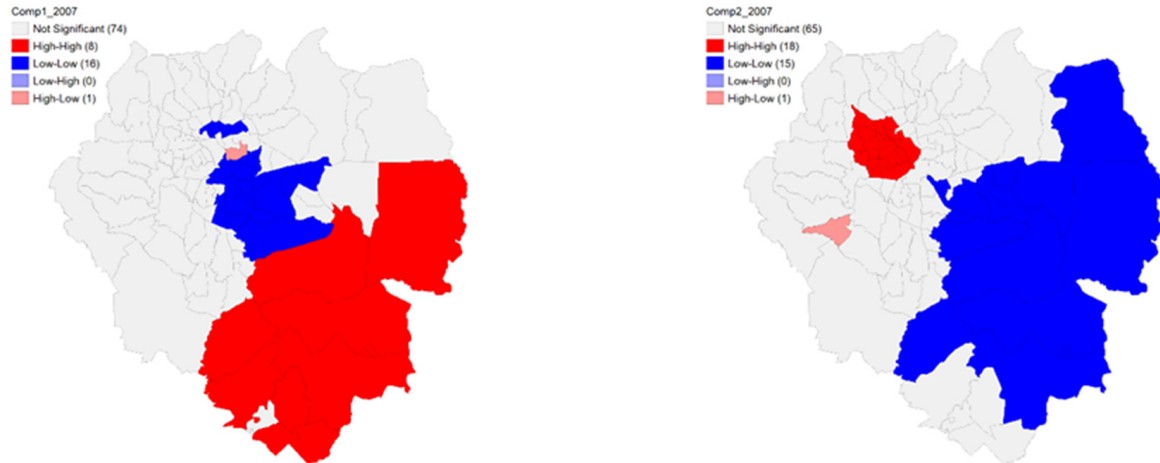

**Figure 5.** LISA cluster map for MDI 2007 component one [LHSS] on the left and component two [HEPSV] on the right.

*MDI 2007 HEPSV (component-2) spatial pattern*

The LISA H-H relationship revealed a high concentration of deprivation in inner-city slum areas. In line with the PCA result for this component, inner-city slums were severely deprived in terms of public health, housing deterioration, and socio-economic vulnerability. Following LISA H-H, the most deprived areas were the entire Addis Ketema sub-city (old CBD), the Lideta sub-city (except for the southwestern part), the western parts of Arada, and the southern parts of the Gulele sub-city. This component showed low deprivation concentration (LISA L-L relation) in the formal settlement-dominated sub-city areas of Yeka, Bole, Chirkos, and Akakai Kaliti. See Figure 5 for a further overview.

*MDI 2007 VUSG (component-3) spatial pattern*

The LISA H-H depicted high deprivation concentrations in peri-urban areas of the Akaki Kaliti sub-city, the inner-city slum areas of the Arada sub-city (except the western strip), and the central part of the Gulele sub-city. Considering the PCA analysis of component 3, the high clustering of deprivation was attributed to the high proportion of migrants who severely lacked electric and bath services, reflecting the character of peri-urban settlements. Furthermore, the high proportion of old dependents as well as the repository of recent migrants reflected the character of the inner-city slum areas. The western neighborhoods of Kolfe-Keranyo and Nifas Silk Sub-cities, as well as strip areas of Bole sub-city, were the low-deprivation concentration areas for VUSG, mainly consisting of formal settlement areas. See Figure 6 for a further overview.

*MDI 2007 PHSC (component-4) spatial pattern*

The LISA H-H showed that deprivation of kitchens and electricity was the main pattern for the undeveloped areas in the peri-urban informal settlement areas of Bole and Akakai Kaliti sub-cities, as well as the inner-city slum areas of the Chirkos sub-city (the central part), Arada sub-city (the western part), and Lideta sub-city (the northern part). On the other hand, the northern suburban areas of Addis Ababa, especially the expansive mixed settlement areas of Gulele and Kolfe-Keranyo sub-cities, were the sites of LISA L-L clustering. See Figure 6 for a further overview.

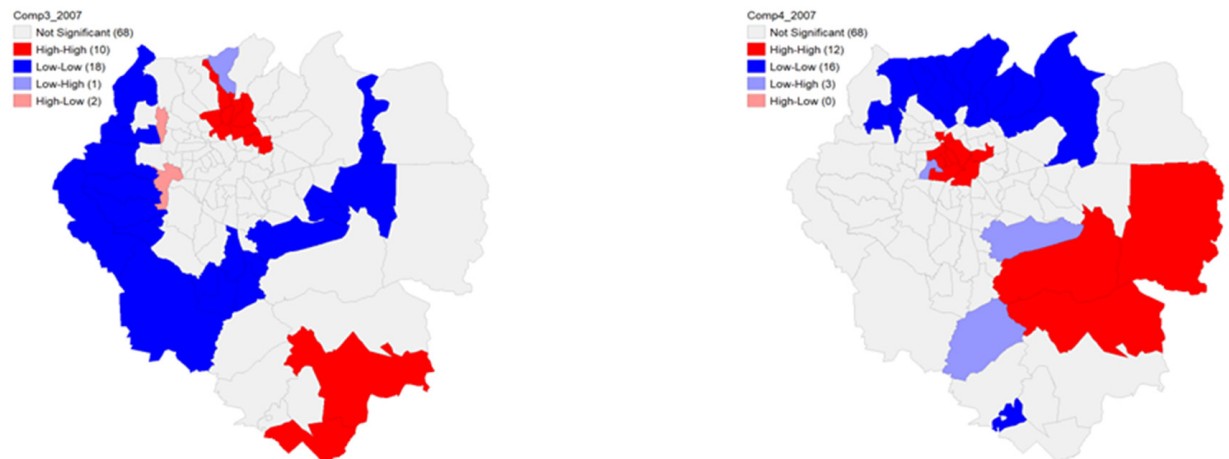

**Figure 6.** LISA cluster map for MDI 2007 component 3 [VUSG] on the left and component 4 [PHSC] on the right.

4.4.2. The Local Spatial Pattern of Deprivations and Inequality for MDI 2016

*MDI 2016 CLVS (component 1) spatial pattern*

The LISA H-H relation for this component depicted a high deprivation concentration in the purely inner-city slum areas of Lideta and Addis Ketema sub-cities. The low-density settlement parts, in the peripheral mixed neighborhood areas of the Kolfe-Keranyo sub-city (west) and the formal settlement areas of the Bole and Yeka sub-cities, showed areas of LISA L-L relation. In conjunction with the PCA result of this component, vulnerable social groups and congested living conditions were characteristics of the inner-city slum, while they were less prevalent in formal neighborhoods. See Figure 7 for a further review.

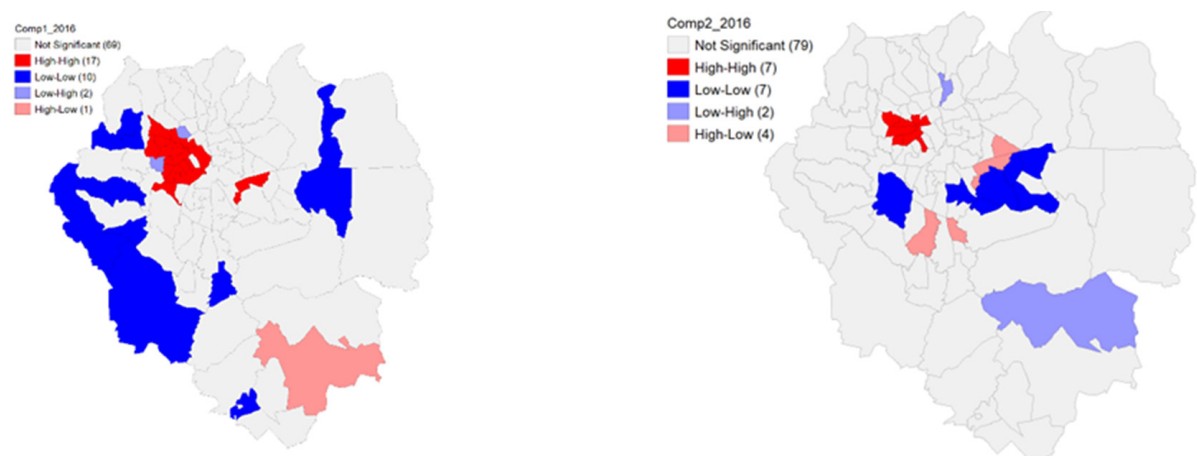

**Figure 7.** LISA cluster map for MDI 2016 component 1 [CLVS] on the left and component 2 [LSES] on the right.

*MDI 2016 LSES (component 2)*

The LISA-H-H showed high deprivation concentrations in the inner-city slum areas of the Addis Ketema sub-city, the northern part of the Lideta sub-city, and the western part of the Arada sub-city. The LISA L-L showed clustering in the formal settlement neighborhoods in the north-central part of the Bole sub-city and a single formal neighborhood for each of the Chirkos and Nifas Silk sub-cities. Concerning the PCA findings of this component, the concentration of deprived human and financial capital assets was one of the features of the inner-city slums, while the intermediate formal areas were concentration areas for dwellers with high socioeconomic status, explaining why poverty was less polarized in formal neighborhoods. See Figure 7 for a further review.

*MDI 2016 EVIN (component 3)*

Based on the LISA H-H relationship, green and road space per capita showed high clustering in the intermediate and suburban areas of the Akaki Kaliti and Bole sub-cities. The aforementioned areas were newly developed formal areas with major road extensions for development; however, they consisted of sparse settlers in 2016. On the contrary, the inner-city slum sub-cities such as the Addis Ketema sub-city, Arada sub-city (the major parts), and Lideta sub-city (the northern part) showed LISA L-L clustering, explaining the lower proportion of green space and road infrastructure per capita for the inner-city slum areas. See Figure 8 for a further overview.

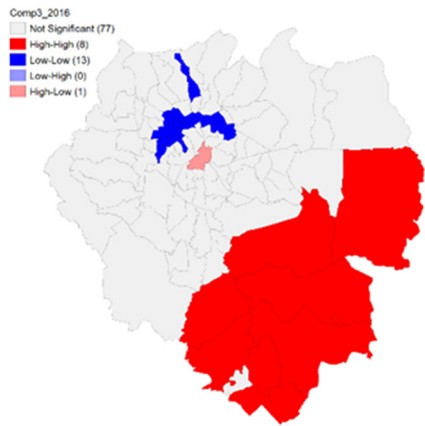

**Figure 8.** LISA cluster map for MDI 2016 component 3 [EVIN].

*4.5. Population Proportion versus MDI*

4.5.1. MDI Classification versus Deprived Population Proportion MDI 2007

The Addis Ketema sub-city had 81.1% of the VHD population, followed by the Lideta sub-city (27.8%). For the three sub-cities (Akaki Kaliti, Kolfe Keranyo, and Gulele), 10–15 percent of the population belongs to the VHD category. In the Akaki Kaliti sub-city, 85.3% of the population belongs to the HD category. HD populations ranged from 19 to 78% in Yeka, Gulele, Nifas Silk, Arada, Chirkos, Kolfe Keranyo, Lideta, and Addis Ketema subcities. Every Addis Ababa sub-city had at least one kebele that belonged to the HD or VHD group. See Figure 9.

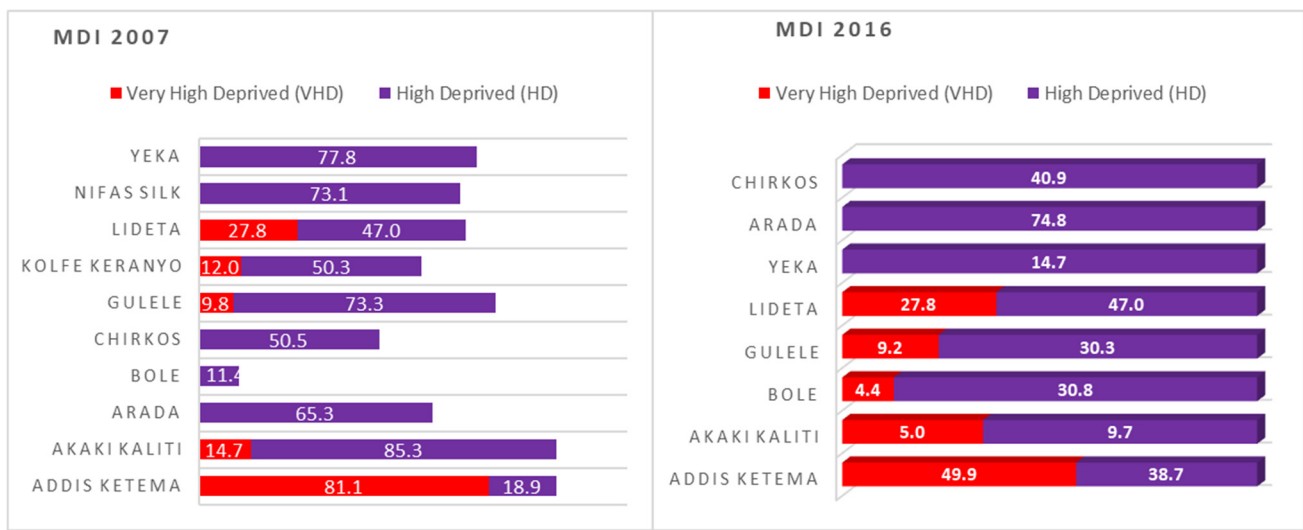

**Figure 9.** Very high-deprived (VHD) and high-deprived (HD) population percent for AA (Addis Ababa) sub-cities based on MDI 2007 and MDI 2016.

### 4.5.2. MDI Classification versus Deprived Population Proportion (MDI 2016)

Sub-cities of Addis Ketema and Lideta accounted for 49.9% and 27.8% of the VHD population, respectively. For Gulele, Akaki Kaliti, and Bole sub-cities, less than 10 percent of the population belongs to the VHD category. The Arada sub-city constituted 74.84% of the HD population. The other sub-cities of Addis Ababa (excluding Nifas Silk and Kolfe Keranyo) comprised 10–47% of the HD population. Sub-cities, such as Kolfe Keranyo and Nifas Silk, do not have a population that belongs to the VHD and HD categories.

### 4.5.3. MDI 2007 and MDI 2016 Comparative Assessment

The most deprived (VHD and HD) population constituted 68.6% and 33.0% of the MDI in 2007 and 2016, respectively. Deprivations have shown declining trends from MDI 2007 to MDI 2016, despite different indicators used for MDI construction. Nonetheless, more research into concentrated poverty, relating deprivation to population density, migration trends, and other factors, is crucial due to the increasing trends in the urbanization of poverty. Compared to MDI 2007, the MDI 2016 population proportion that belongs to HD declined for the sub-cities of Gulele, Akaki Kaliti, Chirkos, and Yeka (sharp decline). The Lideta sub-city population that belongs to HD remained similar in the intervening MDI periods. The Arada sub-city population proportion that belongs to HD increased for MDI 2016 relative to MDI 2007. For MDI 2007, the Kolfe Keranyo sub-city represented 7.1% and 50.3% of the VHD and HD populations, respectively. Nonetheless, for MDI 2016, 0% of the Kolfe Keranyo population has VHD or HD.

### 4.6. The Spatial Trend of Population Density versus MDI

The MDI 2007 and MDI 2016 graphs showed that deprivation peaks in the CBD area and then declines gradually with a noticeable drop in the intermediate city, then rises again until it reaches the equivalent of the CBD peak in the peri-urban area of Addis Ababa and then declines again towards the rural areas.

For MDI 2007, the deprivation score shows a progressive decline until a high peak at 2 km from the CBD, and then again increases to reach a small peak at about 4 km. Then, deprivation showed a recognizable decline at about 5–9 km from the CBD, rose again, roughly equivalent to the CBD in peri-urban areas, and then declined towards the rural areas outside Addis Ababa's peri-urban area. For MDI 2016, the deprivation score showed a sharp decline at 6 km from the CBD, an increase again from 6–7 km from the CBD, and a steady decline from 7–10 km from the CBD. Then, MDI 2016 illustrated a progressive increase to reach a peak equivalent to the CBD around the peri-urban area of the city. Again, MDI 2016 showed a progressive decline in the pure rural area outside Addis Ababa's peri-urban area.

The graph illustrated that the deprivation extent in the inner-city slum is consolidating and expanding since a sharp decline occurred at 2 km from the CBD for the MDI 2007, while the sharp decline was at about 5–6 km from the CBD for the MDI 2016. The population density declined sharply from the CBD, with two recognizable peaks at about 0.5–1 km and 2–4 km and a progressive decline (6–20 km) towards the outskirts of the city for the two MDI periods. The general trends for the two MDI periods revealed that deprivation score and population density had more or less direct correspondence in and around the CBD. However, as one moves from the center of the city to the peri-urban areas, deprivation rises with the progressive drop in population density. The urban form and location, whether it is a formal or informal settlement, influence the spatial concentration of deprivations. The PPMC between MDI 2007 and population density was 0.379. The PPMC between MDI 2016 and population density was 0.497. Based on the findings, deprivation has recently increased in direct proportion to population density, revealing the increasing trends of concentrated poverty in the slums of Addis Ababa (Figure 10).

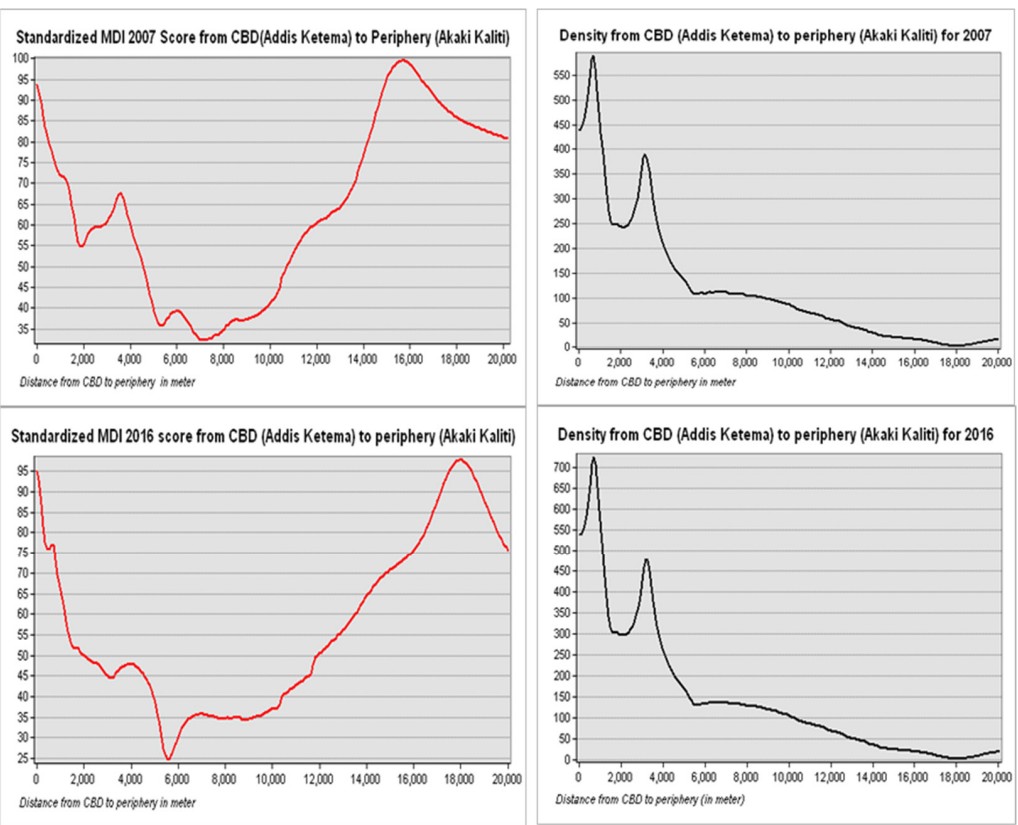

**Figure 10.** The cross-section of deprivation standardized score and population density of Addis Ababa city from the old CBD (Addis Ketema) towards the periphery (Akaki Kaliti sub-city).

## 5. Discussion

### 5.1. Overview of MDI 2007 and MDI 2016: Inductively-Derived PCA Analysis and Spatial Pattern

The MDI 2007 and the MDI 2016 discussed the findings of the first and second research questions in combination by triangulating the pattern of PCA-derived indicators and components with the pattern obtained from the spatial pattern of components. The study also triangulates the findings with SDG, theoretical, and empirical frameworks. For MDI 2007, the result indicated that the first and second components were the main ones, constituting 67.2 percent of the variations in the original dataset, while the third and fourth components explained only 14 percent. For MDI 2016, three of the components explained 60% of the variation in the original data set, with no drastic differences in the percentage of variance explained by the three components. The MDI 2007 and MDI 2016 developments indicate that poverty is multidimensional and multifaceted [61,63,64]. The persistence of low education levels and a high proportion of illiterate HHs as a critical component of the MDI in 2007 and 2016 implied that much work needed to be done to improve informal parental education as well as the overall quality of education to reduce the negative and transgenerational effects of poverty traps [28]. Education level is associated with income, housing inadequacy, unemployment or poorly paid jobs, and poor infrastructure, which rationalizes achieving sustainable urbanization by linking SDG goals 1, 4, 8, and 11. The MDI 2007 and 2016 both revealed a link between vulnerability and deprivation [67–71], implying that the MDI should be used to target resources, green space, and infrastructure to socially excluded and vulnerable groups in order to meet SDGs 1 and 11.

Deprivation is associated with the urban form of Ethiopia, as it is higher in declining inner-city and peri-urban informal settlements while it is lower in intermediate and suburban planned and new development regions [41]. The overlaps of correlated indicators and components over space using PCA and the facts of spatial inequality based

on spatial autocorrelation analysis reinforce the necessity of place-based compensatory policy [25,27,28,44–46]. An area-based policy is justified for Addis Ababa due to a high annual rate of urbanization, a divided city, informal settlement problems, and a housing supply shortage [11,12,25,36,47]. The MDI is helpful for tracking the indicators and extent of problems related to SDG 11's housing inadequacy [14,59], based on the findings of the 2007 MDI.

### 5.1.1. MDI 2007 Component 1 (LHSS): PCA and Spatial Pattern Analysis

This component explained the overlaps and coexistence of a high proportion of substandard housing materials and limited access to services (water, sanitation, waste disposal, cooking, and electricity). In addition, this component also explained inadequate housing facilities and low human capital assets (low educational status, high illiteracy, and a higher number of young dependents). In sum, this component reflected the most pervasive problems common to large parts of Addis Ababa's informal settlements. Because of the dominance of substandard housing, poor services, and facilities, the use of MDI for urban regeneration and prioritization [50,52,53] is rationalized. Therefore, there should be a trend to capture a multitude of problems rather than focusing on some specific criteria not supported by MDI tools, learning from Addis Ababa's slum upgrading experience [48]. The higher proportion of people with lower education and young dependents reflects the typical character of the Ethiopian poor [18]. Yet, the relatively lower electricity provision for this component (LHSS) relative to the second component (HEPSV) indicates that this component reflects the largely suburban and peri-urban informal settlement features. Deprivation is concentrated, based on the LISA H-H relation, in the outskirts, central, and southern suburban informal settlements. Therefore, this component reflected the features of informal settlements outside the inner-city slum.

Yet, the L-L relationship of LISA is depicted in some neighborhoods of the old cultural and education center (piazza) established by the Italian colony (Arada sub-city) and modern formal settlement neighborhood units in the sub-cities of Bole, Chirkos, and Yeka. The SDG 11 housing inadequacy indicators (poor sanitation service, poor water service, and poor housing durability) had a high loading score for this component; therefore, the housing inadequacy indicator of the SDG reflected most of the Addis Ababa informal settlements [29]. In addition to the SDG slum indicators, high-loading score indicators such as low educational capital assets and deficient housing facilities are potential indicators for informal settlements. Because poor education and inadequate housing are overlapping indicators, it makes sense to combine SDGs 4 and 11 for addressing human and physical capital asset deprivation.

### 5.1.2. MDI 2007 Component 2 (HEPSV) PCA Analysis and Spatial Pattern

This component has a strong positive global spatial autocorrelation, explaining that the inner-city slum is an area of concentrated and polarized poverty. This component showed overlaps and coexistence of a high proportion of older housing, deprived private tenure rights (renters), overcrowded living, a dense population, a high proportion of disabilities, and the unemployed. Based on the above fact, this component witnessed the clustering of the most vulnerable dwellers and renters living in aging dwellings. On the contrary, most houses had a high proportion of electricity and waste disposal services [80] due to their inner-city location. The characteristic of a high proportion of old buildings, degenerating housing and infrastructure, and the most vulnerable sub-group, reflects the typical inner-city slum features [23,31,36,74]. The SDG indicators (overcrowding and private tenure owner deprivations) had a higher loading score for this component, which reinforces the inner-city slum nature of this component [23,31,74]. For this component, a high proportion of renters, older houses, and substandard dwellings justify the preponderance of dilapidated and old kebele and municipal rental dwellings in inner-city slums [31].

In addition, the MDI 2007 component 2 (HEPSV) has high positive loading scores for the widowed or divorced FHHs and disabled population proportions. The above result ex-

plained the association of multiple deprivation indicators with the concentration of socially excluded and vulnerable groups [68–70]. Moreover, the concentration of FHH widows or divorcees in the urban core, in line with social organization theory [69], implicates the necessity of gender and marital status mainstreaming interventions. The high proportion of unemployed people in the inner-city slum indicates the economically vulnerable subgroups were the most exposed to multiple deprivations [67]. The spatial pattern based on the LISA map confirmed that the high deprivation area is concentrated in the inner-city slum areas, aligning with the inductively derived and discussed PCA pattern for characterizing inner-city slums. Low-deprivation areas were concentrated in the suburban formal settlement sub-city areas.

### 5.1.3. MDI 2007 Component 3 (VUSG) PCA Analysis and Spatial Pattern

This component revealed the coexistence of vulnerable groups (old dependents, migrants, widowed or divorced FHHs, illiterate), aged buildings, substandard housing walls, less access to electricity, and bathing facilities. This component indicator partly reflects inner-city slum characteristics (old dependents, aged buildings, vulnerable female HHs, and substandard housing materials). In addition, this component indicates characteristics of informal settlements outside the inner-city slum (a high proportion of migrants, a lack of electric services, and substandard housing materials). Similarly, the LISA H-H pattern revealed deprivation clustering in inner-city slums and peri-urban areas. The peri-urban informal or squatter settlements are the main repository for rural-urban migrants [8,31,34,79] and are furnished with little or no electricity. Furthermore, the LISA revealed a high concentration of deprivation in inner-city areas, reflecting the repository of migrants, old buildings, and vulnerable groups in the inner-city areas. The empirical frameworks also mentioned the existence of inner-city squatters in Addis Ababa [31,35], who are mainly migrants. Additionally, a sizable portion of slum residents who resided close to the old CBD rented beds [34] in crowded rooms for temporary and recent migrants.

### 5.1.4. MDI 2007 Component 4 (PHSC) PCA Analysis and Spatial Pattern

This component reflected houses deprived of kitchens in informal settlements due to space shortages for separate kitchens as well as areas deprived of electricity. According to the spatial pattern depicted by the LISA H-H relation, this component showed high deprivation clustering in the inner-city slum areas and the southern and eastern fringe informal settlement areas of Addis Ababa. Yet, the LISA L-L clustering was depicted in the northern and western mixed (formal and informal) settlement areas of Addis Ababa. Based on our 2018 Addis Ababa case study area survey, 60% and 40% of the selected peri-urban and inner-city slum areas do not have kitchens, respectively.

### 5.1.5. MDI 2016 Component 1 (CLVS) PCA Analysis and Spatial Pattern

This component indicated the reinforcing and overlapping nature of deprivations based on indicators of high population density, high building density, widowed or divorced FHHs, older HHs, and unemployed HHs. The first component indicated the overcrowded living conditions of vulnerable sub-groups. The findings implicate that Addis Ababa's inner-city slums are characterized by the area of concentration of the most vulnerable sub-groups living in overcrowded conditions [23,36,74]. Similarly, the spatial pattern of LISA H-H clustering revealed that the vulnerable social groups with high overcrowding were clustered in purely inner-city slum areas. The LISA L-L clustering was depicted in the peripheral mixed-settlement neighborhoods in the western fringe areas and the new formal settlement development areas of Addis Ababa.

### 5.1.6. MDI 2016 Component 2 (LSES) PCA Analysis and Spatial Pattern

The second component of MDI 2016 (LSES) explains how low income reinforces low human capital assets (high proportion of HH illiteracy, low higher education level, and low-paying self-employed jobs). The above finding justifies the link between SDGs 1, 4,

and 8. This is due to education being a means to reduce poverty by increasing people's income [15] and providing employment opportunities. The finding also implies that economic vulnerability reinforces multiple deprivations [65,67]. The PCA findings implicate the rationality of developing MDI from monetary and non-monetary indicators [65,66] since they are mutually reinforcing each other. The LISA H-H revealed how low socio-economic status and deprivation were created due to low human and financial capital assets. Because of the LISA L-L relationship, deprivation showed low clustering in formal neighborhoods, where most dwellers have high socioeconomic status.

### 5.1.7. MDI 2016 Component 3 (EVIN) PCA Analysis and Spatial Pattern

This component indicates that overcrowded places, in terms of population and building density, are less endowed with green and road infrastructure per capita. The above result conformed to the finding that slums in Addis Ababa are places disengaged from infrastructure and public space/greenery [23,36,74]. Yet, the less crowded areas, which are better endowed with green space and road infrastructure, are a feature of Addis Ababa's less developed outskirts. The result also justified the rationality of SDG target 11.7 for providing access to green and public open spaces for vulnerable groups [29]. Additionally, morphological characteristics extracted from satellite images are powerful indicators of deprivation [72,73].

### 5.2. Population Proportion and Density versus Deprivation

In line with the third and fourth research questions, the study analyzes and interprets the proportion of the most deprived population for MDI 2007 and MDI 2016, the correlations and spatial relationships between deprivation and population density, and then suggests the implications for area-based policy. The most deprived populations of Addis Ababa were 68% for MDI 2007 and 33% for MDI 2016. Deprivations have shown declining trends in most sub-cities; however, the inner-city slum sub-cities of Addisketema and Lideta remained in VHD for MDI 2007 and MDI 2016 as well. Similarly, HD also declined in proportion for most sub-cities, except for the inner-city slum sub-cities of Lideta and Arada. For the Kolfe Keranyo sub-city, the most deprived population declined from 62% for MDI 2007 to 0% for MDI 2016. The decline is due to the new, formal-dominated settlement that emerged over the course of the two MDI years.

Other than population proportion, other aspects of urbanization, such as migration and population density, explained concentrated poverty. Regarding density, the PPMC results explicitly indicated that deprivation increased with increasing population density, and the result justified how the density of the poor population definition describes concentrated poverty [37]. In addition, there is a spatial relationship between deprivations, population density, and urban form. For MDI 2007 and MDI 2016, deprivation reaches a peak in the old CBD area, with a progressive decline in the intermediate areas and rising again in the peri-urban area. In the old CBD area, population density increased along with the deprivation score for MDI 2007 and MDI 2016. Based on the spatial profile trends, the high deprivation score and population density near and around the old CBD align with the inner-city radius of 4.5 km from the main CBD [33]. In peri-urban areas, deprivation increased as population density decreased. The spatial profile indicated how deprivation trends varied with the Ethiopian urban form [41], as well as a vivid picture of deprivation polarization and a divided city pattern and the rationales for implementing area-based policy [25]. The threshold area of deprivation in the CBD expanded for MDI 2016, relative to MDI 2007, substantiating the overcrowding and consolidation of the inner-city slum population and density [32,33].

The high deprivation score in the peri-urban area conforms to Alonso's "bid rent model", which suggests that the poor housing and buildings lie in the affordable outskirt area [39]. The increasing deprivation in the fringe area, based on the deprivation profile graph, aligns with Addis Ababa's peri-urban informal settlement characteristics and locations [8,31,34,79]. Even if the old CBD of the inner city's large population concentration

replicated Alonso's "bid rent" model [40], the inner city is a repository of poor populations living in dense quarters. Deprivations increased with increasing rural-urban migrants in peri-urban areas [8,9,34,80], as well as consolidated, partly through bed rent and illegal squatting of migrants, in inner-city slums [34,35]. The findings indicated that the old inner-city slum of Addis Ababa is a concentrated poverty area that conforms to the global north theoretical lens [38]. Nonetheless, with the exception of some areas affected by renewal interventions [76], Addis Ababa's inner-city slum was not affected by the global north's deindustrialization and suburbanization of urban jobs [38].

The relationship between the spatial distribution of deprivation and population size and density rationalized the development of tools for better targeting concentrated poverty. The MDI is rational for better targeting and prioritizing since sector-based pro-poor spending does not fully benefit the vulnerable and poor population, based on the evaluation of 122 World Bank poverty-targeted social programs [42]. Moreover, the high proportion of vulnerable people, including elders, persons with disabilities, females, and the unemployed, implicated a policy framework and tools for better targeting and budget rationing for people needing social protection to realize SDG 1 [29,30]. In summary, a disaggregated area-based welfare policy is an essential intervention [5,44,49,59] to benefit the disadvantaged and vulnerable population and decrease the polarization of poverty.

## 6. Conclusions and Recommendations

Addis Ababa has highly accelerated population growth, largely owing to the high rates of net in-migration from both rural and urban areas, which leads to urban inequality, a divided city, and an escalated housing crisis. If the current high urbanization trend continues, Ethiopia's level of urbanization will constitute more than one-third of the total population before 2035. Nonetheless, urbanization in Ethiopia does not correspond to economic development, which is challenged by environmental hazards, housing inadequacy, poverty, unemployment, and informal settlements. Thus, inclusive cities and sustainable urbanization can be realized through appropriate regularity instruments and tools that tackle concentrated poverty, social exclusion, and spatial inequality.

The research developed the MDI, substantiated the overlaps of problems using indicators, and revealed the spatial pattern of inequality. The study also indicated the disproportional deprivation of large and vulnerable populations by analyzing deprivation versus population size and density. Then, the research suggested the justification for area-based policy based on the findings synthesized in line with the research questions. The overall spatial and statistical analysis indicated the multiple deprivations faced by the urban poor and the deepening spatial and social inequality. MDI tools help to locate and prioritize the poorest of the poor in the endeavor to meet SDG 10 on inequality. Hence, MDI measures the deprivations of sustainable livelihood capital assets suffered by the poor.

By combining PCA indicators and factors with SDGs and conceptual frameworks, the MDI indicators and factors are aligned with the external context to keep track of SDG targets and offer future directions for achieving the SDG. Moran's I and LISA revealed the overall and local spatial pattern of components and enriched the findings of PCA, determining whether the stated problems overlapped in space as well as with the triangulated deductive frameworks. The Morans' I-positive index, based on the coefficients of all components of the MDI in 2007 and 2016, is statistically significant enough to infer concentrated poverty. The LISA spatial pattern indicated a poverty or deprivation polarization between formal and informal settlements. The validation of PCA indicators and components with the LISA pattern revealed spatial inequality as well as the area where deprivation occurred in relation to specific components and indicators. The MDI, a composite index used in the study, is a tool for resource allocation, compensatory policy, and urban upgrading or redevelopment. Hence, the MDI illustrates the many deprivations that the urban poor experience.

In generic form, poor education, income, vulnerability, congestion, non-durable housing, poor services and infrastructure, and poor green access are the critical indicators that overlap in space, explaining the various forms of deprivation faced by the urban poor

between MDI 2007 and 2016. Thus, the above result suggests that integration among most of the SDG's targets should be achieved in tandem to address multiple deprivation issues while also reducing the transgenerational impact of poverty traps. The study suggested additional strong indicators of slums in addition to what is stated in the SDG. For both the MDI 2007 and 2016, low educational attainment and the concentration of vulnerable groups are strong indicators of deprivation in inner-city slum areas.

The most pervasive deprivation problems are deprivations of human capital, substandard housing materials, and poor services, as articulated in MDI 2007 component 1. Therefore, urban regeneration interventions shall be supported by consistent sustainable livelihood capital asset accumulation for the deprived to address most of the SDG's targets in an integrated manner. The triangulation of indicators based on these study findings and the SDG confirmed that Addis Ababa's issues are in sync with globally agreed indicators. For instance, the most pervasive problems, based on MDI 2007 component 1, such as non-durable housing materials, poor sanitation services, and poor water services, conform to the housing inadequacy indicators of SDG 11. The MDI 2007 component 2 indicators of overcrowding and deprivation of own tenure rights (a high proportion of renters) conform to SDG 11 housing inadequacy indicators. In addition to the housing inadequacy indicators aligned with SDG 11, MDI 2007 component 2 reflected general problems of social and economic vulnerability, social exclusion, and poor living conditions. A key strategic intervention is to incrementally upgrade tenure from kebele renter status or no tenure rights to de jure tenure rights, backed with livelihood capital asset accumulation and consolidation. Tenure security is a strategic intervention due to the fact that insecure residents are vulnerable to social exclusion, eviction, and the loss of physical capital asset accumulation.

Based on the assessment of MDI 2016 components 1 and 3, the urban regeneration interventions shall provide adequate open space, green space, and infrastructure access for social, economic, and health-vulnerable groups in slums and informal settlements to meet SDG 11 target 11.7. Furthermore, for the MDI 2016 components, the relationship between indicators of building density, green space, and vulnerable social groups implicates morphological features from satellite images as indicators of deprivation. Based on MDI 2016 component 2, low financial capital and low human capital assets (poor education and low-paying jobs) are reinforcing each other, which justifies the link between SDG 1 on poverty, SDG 4 on education, and SDG 8 on employment opportunity. The above link is justified because the lack of one of the indicators may expose the poor to other forms of deprivation and trigger a vicious circle of poverty. Because the two aspects are mutually reinforcing, MDI development must consider both financial and non-financial components.

Despite Addis Ababa's large number of deprived people in informal settlements, general deprivation trends show a downward trend. Deprivation, however, is polarized in slums and suburban informal settlements. The densely populated inner-city slum sub-cities are in a state of deprivation, exacerbated by inner-city consolidation, illegal building additions, and the repository of recent migrants. In addition, there is an exodus of rural-urban migrants who settled in peri-urban areas despite being faced with sporadic coercive bulldozing. The concentration of deprivation and population density in the inner-city slum, as well as the association of deprivation with less densely populated areas of the peri-urban zone, provided a spatial profile and trend regarding where the disadvantaged people were located. This would help to prioritize area-based targeting of beneficiaries or people who needed social protection or strategic planning interventions. Thus, compensatory budget allocation for small, disaggregated neighborhood units (kebeles) makes the budget reach the poor and vulnerable efficiently and effectively. The justification is that sector-based programs, rather than small area-based targets, obscure the unprivileged, socially excluded, and underclass populations. Furthermore, in a sector-based program, the benefits trickle down to the wealthy and other unintended target groups.

In Ethiopia, the postponement of the planned census in 2018 due to social unrest limited the ability to compare the MDI using similar indicators and forecast future trends in deprivation. Nonetheless, deprivation has been studied recently by researchers interpreting

morphology from cost-effective high-resolution images and sometimes in combination with other socio-economic information. The MDI 2016 development, integrating the survey conducted by the CSA every 5 years with other information, is one option for preparing a cost-effective and time-efficient MDI. The MDI is a composite tool that integrates data from various sources into usable information, including censuses, government surveys, base maps, remote sensing imagery, and so on.

**Supplementary Materials:** The following supporting information can be downloaded at: https://www.mdpi.com/article/10.3390/su15031934/s1, Table S1: AA_census_2007_PCA_Rotated_Matrix and AA_SEPlanning_2016_PCA_Rotated_Matrix. Table S2: AAMDI_2007 and AAMDI_2016 shape file (GIS format). Table S3: MDICensus_2007_SPSS_Principalcomponentanalysis and MDISEPLA_2016_SPSS _Principalcomponentanalysis (in sav file SPSS 20). Table S4: Rotated_Component_Matrix_2007 and Rotated_Component_Matrix_2016 (pdf format). Table S5: MDI 2007 LISA Analysis: MDI 2007 Component 1, 2,3, and 4 cluster, significance, and Moran's I analysis. Table S6: MDI 2016 LISA analysis: MDI 2016 component1,2 and 3 cluster, significance and Moran's I analysis. Table S7: Queen contiguity weight file using geoda software: MDI_2007_Lisweightcomp.gal and MDI_2016_Lisweightcomp.gal. Table S8: MDI2007_deprivationmap and MDI2016_deprivationmap(png format).

**Author Contributions:** Conceptualization, G.B.G.; methodology, G.B.G., S.M.W. and E.G.B.; software, G.B.G.; validation, S.M.W. and E.G.B.; formal analysis, G.B.G.; investigation, G.B.G.; data curation, G.B.G.; writing—original draft preparation, G.B.G.; writing—review and editing, S.M.W. and E.G.B.; visualization, S.M.W.; supervision, S.M.W. and E.G.B.; project administration, G.B.G. All authors have read and agreed to the published version of the manuscript.

**Funding:** This research received no external funding.

**Institutional Review Board Statement:** Not applicable.

**Informed Consent Statement:** Informed consent was obtained from all subjects involved in the study.

**Data Availability Statement:** The 2007 Ethiopian census as well as the household socio-economic expenditure survey was obtained from the Ethiopian Central Statistics Agency (CSA) by writing an official authorized letter from Addis Ababa University to Ethiopian CSA and Addis Ababa City Administration. The data source is Ethiopian CSA and some of the information is from the Addis Ababa City Administration.

**Acknowledgments:** Based on their permission, the authors would like to acknowledge Ethiopia's Central Statistics Authority (CSA) and the Addis Ababa City Administration for providing invaluable data to complete this research.

**Conflicts of Interest:** The authors declare no conflict of interest.

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
