# Peer review of "The Spatial Pattern of Deprivations and Inequalities: The Case of Addis Ababa, Ethiopia"

_sustainability, doi:10.3390/su15031934_

Round 1
Reviewer 1 Report
The Authors' raised an important topic. but the major drawbacks of the paper are word breakdown, describing methodology in the result section, and putting citations in the conclusion which I strongly advise correcting. I put my edits and comments in the text attached.

Author Response
Dear Reviewer, please see the attachment in word format for a point-by-point response
Correspondent author

Reviewer 2 Report
In my opinion, the manuscript is interesting. The authors present an area-based approach to anti-poverty interventions that could be more efficient in reaching the poor and vulnerable than sector-based programs.
However, the manuscript needs some revisions before publishing.
1) There are text formatting problems in the manuscript: many words are unnecessarily hyphenated starting with the word “Ethi-opia” in the title.
2) The way the formulas are written is not consistent with the publisher's standards. Here are some examples:
· Line 404, 405: Subscripts should be used when variables are denoted [as in equation (1)].
· Line 406, 407: Uppercase and lowercase letters are used interchangeably to denote the same quantities.
· Line 465, 466, 467: There are no subscripts in the designations of variables.
The way equation (5) is written is not in line with generally accepted standards for mathematical notation.
3) The word “accordingly”, which occurs in many places in the text, is sometimes abused (for example in lines 11, 29, 38, 100, 151, 415, 472, 762, 832, or 952).
4) It is not clear what the authors mean by “research questions”. They should be better clarified and stated also in the abstract.
Author Response
Dear reviewer, here I attached the point by point response for revision I made in line with your request
Correspondent author

Round 2
Reviewer 2 Report
In my opinion, the manuscript looks much better now but it still requires some minor revisions before publication.
- Line 454: There is a mistake in Equation (1), variables are subscripts.
- The meaning of research questions:
- Line 92 and line 129: "research question answers" – A research question is not an answer; research questions need to be answered.
- Line 135: "This research question's findings" – Findings are needed to answer research questions.
Author Response
The answer for second round revision is attached in pdf format
